# TF-FACE: Time-Frequency Fusion Learning via Frequency-Domain Adaptive and Controllable Enhancement for Trajectory Prediction

**Dongjian Song** [1]  **Yunhao Meng** [1]  **Songjun Huang** [1]  **Jiayi Han** [1]

## Abstract

Accurately predicting the future trajectories of traffic participants is critical for safe, efficient, and human-friendly autonomous driving. Existing learning-based trajectory prediction methods are predominantly time-domain and insufficiently exploit latent frequency information, which limits their capability to capture low-frequency long-term dependencies and high-frequency short-term dynamics. To address this, we propose TF-FACE, a time-frequency learning framework via frequency-domain adaptive and controllable enhancement. TF-FACE introduces a fusion encoder with learnable gated frequency-domain attention that adaptively manipulates band-specific features for trajectory prediction. Building on the fused representation, we design a dual-stage decoder and a band-specific time–frequency dual-consistency loss to enable controllable decoupling and coupling across long- and short-term temporal scales, global and local scales, and then generate final multimodal predictions. Experiments on Argoverse 1 demonstrate that TF-FACE achieves state-of-the-art accuracy, while maintaining real-time inference for autonomous driving. Additional experiments are conducted on Argoverse 2, further validating TF-FACE's performance and generalizability. The source code is publicly available at https://github.com/IMG00180/TF-FACE.

## 1. Introduction

In autonomous driving systems, trajectory prediction of surrounding traffic participants is critical for both conventional hierarchical pipelines and emerging end-to-end architectures (Madjid et al., 2026; Zhou et al., 2025a). Trajectory prediction extends spatial perception of the environment

[1]National Key Laboratory of Automotive Chassis Integration and Bionics, Jilin University, Changchun 130025, China. Correspondence to: Songjun Huang <songjun_huang@jlu.edu.cn>.

*Proceedings of the 43 $^{rd}$ International Conference on Machine Learning*, Seoul, South Korea. PMLR 306, 2026. Copyright 2026 by the author(s).

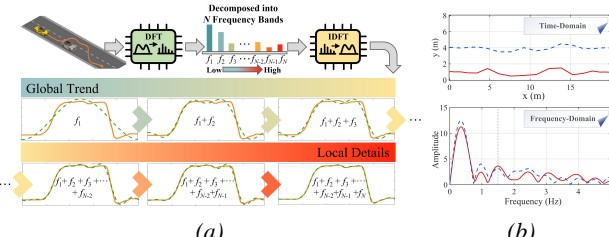

*Figure 1.* Time–frequency characteristics of agent trajectories. (a) Progressive reconstruction of a time-domain overtaking trajectory. (b) Differences between two trajectories that are similar in the time domain but differ in the frequency domain.

into the temporal dimension, providing richer information for situation understanding and decision-making, and is necessary for safe, efficient, and human-friendly autonomous driving (Zhang & Li, 2022).

As autonomous driving systems increasingly demand higher prediction accuracy and longer forecasting horizons, data-driven learning-based approaches have received increasing attention (Li et al., 2019). Most existing learning-based trajectory prediction methods operate primarily in time-domain (Zhou et al., 2022c; Feng et al., 2023; Liu et al., 2024), which extract interaction and coupling patterns from temporal observations of traffic scenes and learn complex nonlinear transitions from historical motion to future trajectories, producing unimodal or multimodal forecasts for each agent (Gu et al., 2021; Zhang et al., 2024b).

However, a trajectory in time-domain can also be viewed as a combination of various signals at different frequencies (Zhang et al., 2023). After transforming a trajectory into frequency-domain, local point-wise temporal correlations are mapped into a set of frequency components with a global perspective, revealing latent structures that are difficult to access from raw temporal signals. As illustrated in Figure 1a, applying the Discrete Fourier Transform (DFT) to an overtaking trajectory produces $N$ frequency components $f_1, f_2, \ldots, f_N$, ordered from low to high frequencies. Reconstructing the trajectory from these frequency components via the Inverse DFT (IDFT) shows that by only using low-frequency components, the reconstructed trajectory can restore the basic trend of the original trajectory. As higher-frequency components are injected, the reconstruction gradually exhibits finer-grained motion details and becomes closer to the original trajectory. This suggests a complementary role of different bands: low-frequency

components capture the global motion trend, whereas high-frequency components characterize local motion details. As shown in Figure 1b, two trajectories that are highly similar in time-domain can be notably different after being transformed into frequency-domain. Since both Trajectory 1 and 2 represent lane-keeping, their low-frequency components, which reflect global trends, are generally consistent, with only minor differences at a few peaks. By contrast, their high-frequency components differ significantly, suggesting local-detail differences that are difficult to reveal from time-domain signals alone.

These observations suggest there are limitations in purely time-domain learning-based trajectory prediction. First, low-frequency components that encode global trends and high-frequency components that encode local details are entangled in the raw temporal signal (Zhang et al., 2023; Yang et al., 2022). Time-domain models may struggle to explicitly disentangle and fully exploit long-term dependencies and global motion trends (Chen et al., 2025). In particular, recurrent architectures often suffer from vanishing or exploding gradients when modeling long sequences. Alternatively, enforcing global dependencies by stacking deep spatiotemporal attention blocks or enlarging receptive fields may introduce information mixing and unnecessary computational overhead. Second, due to the spectral bias of deep neural networks, models typically fit low-frequency components earlier and learn high-frequency components more slowly or even under-emphasize them (Xu & Zhou, 2021). Without explicit guidance on high-frequency information, time-domain approaches may fail to capture transient motion details and may also struggle to distinguish informative high-frequency patterns from perturbations, resulting in the accumulation of local prediction errors in dynamic scenes, leading to degraded prediction accuracy (Feng et al., 2023; Liu et al., 2024; Zhou et al., 2022c).

Motivated by these observations, it is necessary to go beyond purely time-domain learning and incorporate frequency-domain information. Frequency-domain learning has been widely used in signal processing and pattern recognition. In autonomous driving, however, only a few studies have explored frequency-domain learning for environment perception, trajectory prediction, and decision-making and planning. Most existing efforts target a single frequency band or a specific module, leaving key problems underexplored, including high–low frequency decoupling with cross-scale collaboration, joint modeling of global trajectory trends and local motion details, and principled time–frequency consistency constraints during training.

To address the limitations of time-domain learning for trajectory prediction and the gap in frequency-domain learning in autonomous driving, we propose TF-FACE, a time–frequency learning framework with frequency-domain adaptive and controllable enhancement (Figure 2). TF-FACE consists of Frequency-domain Adaptive Fusion Encoding Module (FAE), Preliminary Trajectory Decoding

Module (PreD, the first decoding stage), Low-Frequency Global Trend Feature Extraction Module (LFE), High-Frequency Local Feature Extraction Module (HFE), and Dual-Branch Parallel Decoding Module (DuaD, the second decoding stage). First, FAE encodes agents' historical trajectories, vectorized map, and relative spatial poses, and then performs multi-head cross-attention to fuse these representations, producing an agent–map fused feature and an agent–map–interaction fused feature. Next, PreD decodes the agent–map fused feature into preliminary trajectories via an MLP. Then, LFE transforms the preliminary trajectories into the frequency-domain using DFT and applies frequency-domain MLP (freMLP) to extract reliable future low-frequency global features that capture motion trends, which are converted back to the time-domain via IDFT. In parallel, HFE applies a high-pass filter to frequency-domain historical trajectories to isolate high-frequency components, converts them back to the time-domain via IDFT, and uses an MLP to obtain historical high-frequency local features. Finally, DuaD takes future global features and historical local features as inputs to two branches, respectively. Each branch further fuses its inputs with the agent–map fused feature and the agent–map–interaction fused feature through cross-attention, and then decodes multimodal global trend trajectories and local detail trajectories via two MLP heads. Their combination forms the final multimodal predictions.

Overall, the main contributions are summarized as follows: (1) A new time–frequency fusion learning framework, TF-FACE, is proposed for multimodal trajectory prediction, enabling adaptive and controllable exploitation of both low- and high-frequency information to improve prediction performance. To the best of our knowledge, TF-FACE is the first autonomous-driving trajectory prediction model that leverages frequency-domain learning in a systematic end-to-end manner, spanning scene encoding, feature extraction, and trajectory decoding. (2) On the encoding side, a learnable gated frequency-domain attention mechanism is introduced to adaptively enhance or suppress different frequency bands for the prediction task, yielding a collaborative representation of long-term dependencies and transient motion patterns while effectively mitigating high-frequency noise perturbations. (3) On the decoding side, a controllable band-wise feature extraction strategy and a frequency-aware two-stage decoder are developed, together with a carefully designed band-specific time–frequency dual consistency loss, enabling cross-scale, controllable decoupling and coupling between low and high frequencies, as well as between global trends and local details, providing reliable trend guidance and detail completion for trajectory prediction.

## 2. Related work

### 2.1. Time-Domain Trajectory Prediction

Time-domain learning has long been the dominant paradigm for autonomous driving trajectory prediction, and existing

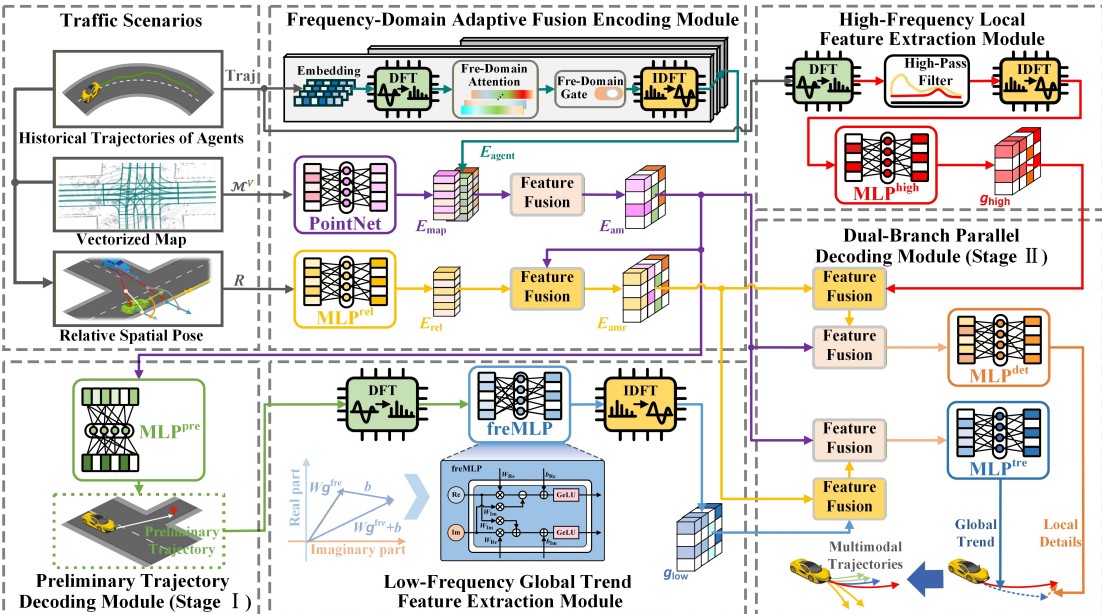

*Figure 2.* Overview of the proposed TF-FACE architecture.

approaches can be broadly categorized into four families: recurrent sequence modeling, convolution- or graph-based modeling of spatial interaction features, Transformer-based modeling of global dependencies, and generative modeling of trajectories. Early work commonly employed recurrent architectures such as LSTMs and GRUs to encode historical trajectories, primarily capturing short-term motion continuity (Xie et al., 2020). Subsequent research shifted toward structured scene representations and graph learning. Methods such as LaneGCN (Liang et al., 2020), VectorNet (Gao et al., 2020) and SocialCircle (Wong et al., 2024), leverage convolutional networks, graph neural networks, or social interaction modeling to improve the representation of multi-agent spatial interactions in structured driving scenes. More recently, Transformers and their variants have been widely adopted for trajectory prediction (Feng et al., 2023; Liu et al., 2024), using attention mechanisms to capture long-term dependencies and complex interaction patterns, leading to substantial progress. For example, Scene Transformer (Ngiam et al., 2022) jointly models multiple agents and scene elements within a unified attention framework, while HiVT (Zhou et al., 2022c) and Wayformer (Nayakanti et al., 2023) further enhance scene encoding capacity and scalability through hierarchical interaction decomposition and more efficient attention designs. In parallel, generative models have become increasingly popular. By learning the multimodal joint distribution over historical context and future trajectories, they naturally produce diverse predictions spanning different behavior modes. Trajectron++ (Salzmann et al., 2020) adopts conditional variational generative modeling with graph structures to explicitly capture multi-agent interactions and output probabilistic future trajectories. Diffusion-based approaches, such as MotionDiffuser (Jiang et al., 2023), Diff-Refiner (Zhou et al., 2025b) and DiffRefiner (Yin et al., 2026), represent and sample the future

trajectory distribution through iterative denoising, yielding richer and more diverse forecasts.

Despite these advances, most trajectory prediction methods are built upon purely time-domain representations, implicitly assuming that future motion is primarily governed by temporal relations. As a result, they rely heavily on deep networks to learn dynamics in the time-domain without explicitly modeling frequency-domain information. This limits the ability to disentangle and jointly exploit long-term dependencies and short-term motion patterns, making long-horizon modeling prone to vanishing or exploding gradients and reducing sensitivity to non-stationary motion details. Moreover, without adaptive regulation of band-wise importance, time-domain predictors often struggle to separate noise-like perturbations that manifest as high-frequency components, which can further undermine robustness.

### 2.2. Frequency-Domain Time Series Prediction

Existing frequency-domain learning methods for time-series forecasting can be broadly grouped into two categories. The first category is frequency-only approaches (Yi et al., 2023; Wang et al., 2025), which focus on frequency-domain decomposition and explicitly exploit the compressibility and separability of spectra for representation learning (Rao et al., 2021). For example, FEDformer (Zhou et al., 2022b) leverages compact spectral representations induced by frequency decomposition and captures long-term dependencies and global structures using only a small number of key frequency components. The second category is time–frequency fusion approaches, which introduce an auxiliary frequency-domain branch into a time-domain backbone and fuse frequency-domain features with time-domain representations. For instance, FiLM (Zhou et al., 2022a) compresses historical time-domain information via Legendre polynomial projec-

tions and then applies Fourier projections to enhance features in the frequency-domain, improving representation quality for long sequences.

In autonomous driving, only a limited number of studies have recently begun to incorporate frequency-domain learning. View Vertically (Wong et al., 2022) constructs hierarchical representations from a frequency-domain perspective to characterize pedestrian motion patterns at different scales. PatchTraj (Liu et al., 2025) treats frequency-domain components as a complement to time-domain modeling to enhance the representation of pedestrian motion features. DiffWT (Chen et al., 2024) performs time–frequency decomposition through wavelet transforms and embeds frequency-domain features into a diffusion-based generative framework to model non-stationary pedestrian trajectory variations at a finer granularity. These explorations demonstrate the effectiveness of incorporating frequency-domain information into autonomous driving tasks; however, they are often restricted to a single frequency band or a specific module, and a comprehensive frequency-domain learning paradigm remains unexplored, particularly for autonomous driving trajectory prediction.

## 3. Methodology

### 3.1. Problem Formulation

Trajectory prediction can be formulated as forecasting multimodal future trajectories and their corresponding mode probabilities for multiple target agents, conditioned on the observed agent histories and map context. Formally, consider a scene with $N_a$ agents (i.e., $N_a$ prediction targets). Let $\mathbf{X} = \{\mathbf{x}_1, \mathbf{x}_2, \ldots, \mathbf{x}_{N_a}\}$ denote the set of historical observations, where $\mathbf{x}_i = \{\mathbf{x}_{i,t_0-H+1}, \mathbf{x}_{i,t_0-H+2}, \ldots, \mathbf{x}_{i,t_0}\}$, $i \in \{1, 2, \ldots, N_a\}$ and each state is $\mathbf{x}_{i,t} = [x_{i,t}, y_{i,t}, \mathbf{v}_{i,t}]$, $t \in [t_0 - H + 1, t_0]$. Here, $H$ is the history length and $t_0$ is the current time step. $x_{i,t}$, $y_{i,t}$ and $\mathbf{v}_{i,t}$ denote the longitudinal position, lateral position, and velocity vector of agent $i$ at time $t$, respectively. For each target agent $i$, all agent positions are expressed in a local coordinate frame centered at agent $i$'s position at time $t_0$. The map context is denoted by $\mathcal{M}$, which includes lane boundaries, lane centerlines, intersection geometry, traffic signals, and other road semantic attributes.

The task is to generate a set of plausible future trajectories for all $N_a$ agents, $\mathbf{Y} = \{\mathbf{y}_1, \mathbf{y}_2, \ldots, \mathbf{y}_{N_a}\}$, where for agent $i$ $\mathbf{y}_i = \{\mathbf{y}_i^1, \mathbf{y}_i^2, \ldots, \mathbf{y}_i^K\}$, $i \in \{1, 2, \ldots, N_a\}$, and the $k$-th mode is $\mathbf{y}_i^k = \{\mathbf{y}_{i,t_0+1}^k, \mathbf{y}_{i,t_0+2}^k, \ldots, \mathbf{y}_{i,t_0+F}^k\}$, $k \in \{1, \ldots, K\}$, with $\mathbf{y}_{i,t}^k = [x_{i,t}, y_{i,t}]$, $t \in [t_0+1, t_0+F]$. Here $K$ is the number of trajectory modes and $F$ is the prediction horizon. The model also outputs the corresponding mode probabilities $\boldsymbol{\Omega} = \{\boldsymbol{\omega}_1, \boldsymbol{\omega}_2, \ldots, \boldsymbol{\omega}_{N_a}\}$, where $\boldsymbol{\omega}_i = \{\omega_i^1, \omega_i^2, \ldots, \omega_i^K\}$ is the probability vector over modes for agent $i$. Accordingly, the multimodal prediction task for any target agent $i$ can be written as estimating

the following conditional distribution:

$$P\left(\mathbf{y}_i \mid \mathbf{X}, \mathcal{M}\right) = \sum_{k=1}^{K} \omega_i^k \cdot P\left(\mathbf{y}_i^k \mid \mathbf{X}, \mathcal{M}\right). \quad (1)$$

The raw agent observations and map context are further processed into three types of input features: the agents' time-domain historical trajectories $\mathbf{Traj} \in \mathbb{R}^{N_a \times 2 \times H}$, the vectorized map $\mathcal{M}^{\mathrm{V}} \in \mathbb{R}^{N_m \times B \times d_m}$, and the relative spatial pose matrix $\boldsymbol{R} \in \mathbb{R}^{(N_a+N_m) \times (N_a+N_m) \times 5}$. The detailed construction of these features is provided in Appendix A.1. Accordingly, Eq. 1 can be written more explicitly as:

$$P\left(\mathbf{y}_i \mid \mathbf{X}, \mathcal{M}\right) = \sum_{k=1}^{K} \omega_i^k P\left(\mathbf{y}_i^k \mid \mathbf{Traj}, \mathcal{M}^{\mathrm{V}}, \boldsymbol{R}\right). \quad (2)$$

### 3.2. Frequency-domain Adaptive Fusion Encoding Module (FAE)

FAE adopts different encoders to obtain the embedding of the agents' $\mathbf{Traj}$, denoted as $\boldsymbol{E}_{\mathrm{agent}}$, the embedding of the vectorized map $\mathcal{M}^{\mathrm{V}}$, denoted as $\boldsymbol{E}_{\mathrm{map}}$, and the embedding of the relative spatial pose matrix $\boldsymbol{R}$, denoted as $\boldsymbol{E}_{\mathrm{rel}}$. The detailed encoding procedures for $\mathcal{M}^{\mathrm{V}}$ and $\boldsymbol{R}$ are provided in Appendices A.2 and A.3, respectively. The encoding of $\mathbf{Traj}$ is described below.

To fully extract full-band features from $\mathbf{Traj}$, as shown in Figure 3, a Gated Frequency-Domain Attention Mechanism (GfreAttn) is developed under a Feature Pyramid Network (FPN). FPN progressively maps features from shallow to deep layers, enabling multi-temporal-scale representation of global trends and local details. Meanwhile, GfreAttn adaptively reweights different frequency bands, alleviating the systematic attenuation of high-frequency components accumulated in deeper layers. In this way, low-frequency trends and high-frequency details are jointly encoded in a coupled manner. Specifically, the FPN is constructed with three stacked levels: the shallow level takes the raw time-domain $\mathbf{Traj}$ as input, the middle level takes the shallow-level output as input, and the deep level takes the middle-level output as input. Without loss of generality, the first level is used as an example to illustrate the workflow of FPN and GfreAttn.

As shown in Figure 3, $\mathbf{Traj}$ is first projected to a higher-dimensional feature space using a 1D residual convolution block (Res1d). Res1d consists of two 1D convolution layers with a residual connection and uses the GeLU activation. It preserves the original information in $\mathbf{Traj}$ via the residual pathway while enhancing the capture of local motion details through convolution:

$$\widetilde{\mathbf{Traj}} = \mathrm{Res1d}(\mathbf{Traj}), \quad (3)$$

where $\widetilde{\mathbf{Traj}} \in \mathbb{R}^{N_a \times 32 \times H}$ denotes the time-domain historical trajectory features after detail enhancement.

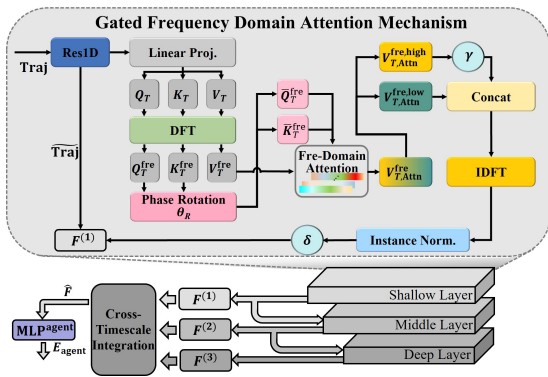

*Figure 3.* Agent historical trajectory encoding network based on FPN and GfreAttn.

Next, in GfreAttn, **Traj** is first linearly projected to obtain a more expressive hidden representation with richer semantics. The projected features are then transformed from time-domain to frequency-domain via DFT. A learnable channel-wise phase rotation vector $\boldsymbol{\theta}_R \in \mathbb{R}^{32}$ is further introduced to align spectral components across frequency bands, after which frequency-domain self-attention is applied to adaptively enhance or suppress full-band information:

$$\widehat{\boldsymbol{Q}}_T^{\text{fre}} = \boldsymbol{Q}_T^{\text{fre}} \cdot e^{j(\boldsymbol{\theta}_R \otimes \boldsymbol{f})}, \quad \widehat{\boldsymbol{K}}_T^{\text{fre}} = \boldsymbol{K}_T^{\text{fre}} \cdot e^{-j(\boldsymbol{\theta}_R \otimes \boldsymbol{f})}, \quad (4)$$

$$\boldsymbol{V}_{T,\text{Attn}}^{\text{fre}} = \text{softmax}\left( \text{Re}\left( \widehat{\boldsymbol{Q}}_T^{\text{fre}} \cdot (\widehat{\boldsymbol{K}}_T^{\text{fre}})^* \right) \right) \cdot \boldsymbol{V}_T^{\text{fre}}. \quad (5)$$

In Eqs. 4 and 5, $\boldsymbol{Q}_T^{\text{fre}}$, $\boldsymbol{K}_T^{\text{fre}}$ and $\boldsymbol{V}_T^{\text{fre}} \in \mathbb{C}^{N_a \times 32 \times H}$ denote the complex-valued frequency-domain query, key, and value, encoding both magnitude and phase information across frequency bands, and can be derived by $\text{DFT}(\boldsymbol{Q}_T)$, $\text{DFT}(\boldsymbol{K}_T)$, and $\text{DFT}(\boldsymbol{V}_T)$, respectively. $\boldsymbol{Q}_T$, $\boldsymbol{K}_T$ and $\boldsymbol{V}_T \in \mathbb{R}^{N_a \times 32 \times H}$ are the time-domain query, key, and value, and are computed as $\boldsymbol{W}_T^Q \cdot \widetilde{\textbf{Traj}}$, $\boldsymbol{W}_T^K \cdot \widetilde{\textbf{Traj}}$, and $\boldsymbol{W}_T^V \cdot \widetilde{\textbf{Traj}}$, respectively. $\boldsymbol{W}_T^Q$, $\boldsymbol{W}_T^K$ and $\boldsymbol{W}_T^V \in \mathbb{R}^{32 \times 32}$ are learnable linear projection matrices. $\widehat{\boldsymbol{Q}}_T^{\text{fre}}$ and $\widehat{\boldsymbol{K}}_T^{\text{fre}} \in \mathbb{C}^{N_a \times 32 \times H}$ are the phase-rotated frequency-domain query and key. $j$ is the imaginary unit with $j^2 = -1$, $\boldsymbol{f} \in \mathbb{R}^H$ is the frequency index vector corresponding to DFT bins and $\otimes$ denotes broadcasted multiplication. $(\cdot)^*$ denotes complex conjugation, and $\text{Re}(\cdot)$ takes the real part. $\boldsymbol{V}_{T,\text{Attn}}^{\text{fre}} \in \mathbb{C}^{N_a \times 32 \times H}$ is the frequency-domain value after attention weighting, which aggregates coupled information over all frequency bands.

During the hierarchical stacking of multi-scale temporal convolutions and frequency-domain attention in FPN, high-frequency components tend to be progressively attenuated due to convolutional smoothing. To alleviate this issue, GfreAttn applies selective enhancement to the high-frequency components. Specifically, $V_{T,\text{Attn}}^{\text{fre}}$ is explicitly split along the frequency dimension into a low-frequency part $V_{T,\text{Attn}}^{\text{fre,low}} \in \mathbb{C}^{N_a \times 32 \times H_{\text{low}}}$ and a high-frequency part $V_{T,\text{Attn}}^{\text{fre,high}} \in \mathbb{C}^{N_a \times 32 \times H_{\text{high}}}$, where $H_{\text{low}} + H_{\text{high}} = H$ and

$H_{\text{low}} : H_{\text{high}}$ follows a predefined split ratio. A learnable channel-wise gating factor $\gamma$ is introduced for the high-frequency part, and the high-frequency magnitude is modulated via channel-wise multiplication, yielding $\gamma \otimes \boldsymbol{V}_{T,\text{Attn}}^{\text{fre,high}}$. This allows GfreAttn to adaptively adjust the contribution of high-frequency components according to task demands. The split ratio may vary across benchmarks.

The enhanced high-frequency components are concatenated with the low-frequency components and transformed back to time-domain via IDFT, restoring the temporal length to $H$. In addition, to mitigate the interference caused by scale mismatch across frequency bands, Instance Normalization (IN) is applied to reduce distribution discrepancies across channels. The normalized features are then fused with the original time-domain features using a residual scaling parameter $\delta$, yielding a time–frequency representation with frequency-domain enhancement:

$$\boldsymbol{F}^{(1)} = \widetilde{\textbf{Traj}} + \delta \cdot \text{IN}\Bigg( \text{IDFT}\Big( \text{Concat}\big($$
$$\boldsymbol{V}_{T,\text{Attn}}^{\text{fre,low}}, \gamma \boldsymbol{V}_{T,\text{Attn}}^{\text{fre,high}}\big)\Big)\Bigg), \quad (6)$$

where $\boldsymbol{F}^{(1)} \in \mathbb{R}^{N_a \times 32 \times H}$ is the shallow layer output of FPN, i.e., the frequency-adaptively enhanced time–frequency semantic feature. As shown in Figure 3, the shallow output $\boldsymbol{F}^{(1)}$ is fed into the middle layer to produce $\boldsymbol{F}^{(2)} \in \mathbb{R}^{N_a \times 64 \times H/2}$, which is further passed to the deep layer to obtain $\boldsymbol{F}^{(3)} \in \mathbb{R}^{N_a \times 128 \times H/4}$. The computations in the middle and deep layers follow the same procedure as Eqs. 3-6. Through multi-temporal-scale feature extraction in FPN, the trajectory length is progressively compressed while feature channels are expanded.

Next, the multi-level semantic features $\boldsymbol{F}^{(1)}$, $\boldsymbol{F}^{(2)}$ and $\boldsymbol{F}^{(3)}$ are integrated across temporal scales to obtain the time–frequency fused history representation $\widehat{\boldsymbol{F}} \in \mathbb{R}^{N_a \times D \times H}$ (see Appendix A.4 for details). To improve the stability of semantic representations, the last time-step feature $\widehat{\boldsymbol{F}}_{:,:,\text{end}}$ is extracted and projected by $\text{MLP}^{\text{agent}}$ to obtain the final historical-trajectory embedding $\boldsymbol{E}_{\text{agent}} \in \mathbb{R}^{N_a \times D}$ for each agent containing in current scenario.

After obtaining the embeddings of the three input feature types, Multi-Head Attention (MHA) is used to fuse $\boldsymbol{E}_{\text{agent}}$ with the vectorized map embedding $\boldsymbol{E}_{\text{map}}$, yielding a scene-level global semantic feature $\boldsymbol{E}_{\text{am}} \in \mathbb{R}^{N_a \times D}$ that captures the global trend of future trajectories. To further preserve local interaction details, MHA is applied again to fuse $\boldsymbol{E}_{\text{am}}$ with the relative-pose embedding $\boldsymbol{E}_{\text{rel}}$, producing an agent-level detailed semantic feature $\boldsymbol{E}_{\text{amr}} \in \mathbb{R}^{N_a \times D}$ that contains the information required for decoding.

### 3.3. Preliminary Trajectory Decoding Module (PreD, the First Decoding Stage)

PreD serves as the first stage of the proposed two-stage decoding pipeline, and it aims to produce an initial forecast of future trajectories. Although PreD functions as a trajectory decoder, it is conceptually closer to a task-driven re-encoding of the outputs from FAE, where trajectory supervision directly constrains the learned time–frequency representations. Specifically, PreD decodes the scene-level global semantic feature $\boldsymbol{E}_{\mathrm{am}}$ using $\mathrm{MLP}^{\mathrm{pre}}$ to extract a coarse future motion pattern for each target agent, yielding a single-modal preliminary trajectory $\mathbf{Y}_{\mathrm{pre}} \in \mathbb{R}^{N_a \times 2 \times F}$.

### 3.4. Low-Frequency Global Trend Feature Extraction Module (LFE) and High-Frequency Local Feature Extraction Module (HFE)

Agent trajectories typically exhibit smooth, persistent, and often quasi-periodic evolution over long horizons. As a result, the low-frequency components are relatively stable and less affected by short-term perturbations. Therefore, the global motion trend reflected in $\mathbf{Y}_{\mathrm{pre}}$ over the prediction horizon $F$ (i.e., its low-frequency components) can be regarded as reliable. Meanwhile, trajectories also contain high-frequency components that fluctuate more strongly and are highly sensitive to transient disturbances. Without explicitly decoupling the high-frequency components and when the full-band coupled signal is only processed through simple nonlinear transformations, the PreD module struggles to learn fine-grained motion-detail variations. This often leads to large errors in local details of $\mathbf{Y}_{\mathrm{pre}}$, which can accumulate over time. Consequently, we adopt different feature extraction strategies for low- and high-frequency components to enable controllable decoupled utilization of trend and detail information.

**LFE:** Reliable low-frequency features are extracted from $\mathbf{Y}_{\mathrm{pre}}$ to capture future motion trends and provide trend guidance for DuaD. Specifically, $\mathbf{Y}_{\mathrm{pre}}$ is first transformed into the frequency-domain via DFT, yielding $\mathbf{Y}_{\mathrm{pre}}^{\mathrm{fre}}$. freMLP is then applied to $\mathbf{Y}_{\mathrm{pre}}^{\mathrm{fre}}$ to extract the future low-frequency global features $\boldsymbol{g}_{\mathrm{low}}^{\mathrm{fre}} \in \mathbb{C}^{N_a \times D \times F}$, which are mapped back to the time-domain via IDFT and are then temporally pooled to obtain reliable low-frequency features $\boldsymbol{g}_{\mathrm{low}} \in \mathbb{R}^{N_a \times D}$. The computation of freMLP is given by:

$$\mathrm{Re}\left(\boldsymbol{g}_{\mathrm{low}}^{\mathrm{fre}}\right) = \mathrm{GeLU}\left(\boldsymbol{W}_{\mathrm{Re}} \cdot \mathrm{Re}\left(\mathbf{Y}_{\mathrm{pre}}^{\mathrm{fre}}\right) - \right.$$
$$\left. \boldsymbol{W}_{\mathrm{Im}} \cdot \mathrm{Im}\left(\mathbf{Y}_{\mathrm{pre}}^{\mathrm{fre}}\right) + \boldsymbol{b}_{\mathrm{Re}}\right), \quad (7)$$

$$\mathrm{Im}\left(\boldsymbol{g}_{\mathrm{low}}^{\mathrm{fre}}\right) = \mathrm{GeLU}\left(\boldsymbol{W}_{\mathrm{Im}} \cdot \mathrm{Re}\left(\mathbf{Y}_{\mathrm{pre}}^{\mathrm{fre}}\right) + \right.$$
$$\left. \boldsymbol{W}_{\mathrm{Re}} \cdot \mathrm{Im}\left(\mathbf{Y}_{\mathrm{pre}}^{\mathrm{fre}}\right) + \boldsymbol{b}_{\mathrm{Im}}\right), \quad (8)$$

where $\boldsymbol{W}_{\mathrm{Re}}$ and $\boldsymbol{W}_{\mathrm{Im}} \in \mathbb{R}^{D \times 2}$ are the learnable weight matrices for the real and imaginary parts of freMLP, and $\boldsymbol{b}_{\mathrm{Re}}$ and $\boldsymbol{b}_{\mathrm{Im}} \in \mathbb{R}^{1 \times 1 \times D}$ are the corresponding learnable biases.

freMLP explicitly extracts and represents the reliable low-frequency component $\boldsymbol{g}_{\mathrm{low}}$ from $\mathbf{Y}_{\mathrm{pre}}$, leading to a more concentrated low-frequency energy distribution.

**HFE:** Since the future high-frequency components embedded in $\mathbf{Y}_{\mathrm{pre}}$ cannot be directly used for trajectory prediction, a more robust strategy is to extract high-frequency features from the observed history. Although historical high-frequency cues are not as informative as true future high-frequency signals, they are much easier to obtain and are generally more stable and remain meaningful due to the temporal continuity of local motion patterns. As shown in Figure 3, a high-pass filter (HPF) is applied to the agents' frequency-domain historical trajectories to isolate high-frequency components according to a predefined ratio. The filtered signal is then transformed back to the time-domain via IDFT and encoded by $\mathrm{MLP}^{\mathrm{high}}$ to obtain the historical high-frequency local feature $\boldsymbol{g}_{\mathrm{high}} \in \mathbb{R}^{N_a \times D}$.

$$\boldsymbol{g}_{\mathrm{high}} = \mathrm{MLP}^{\mathrm{high}}(\mathrm{IDFT}(\mathrm{HPF}(\mathrm{DFT}(\mathbf{Traj})))). \quad (9)$$

### 3.5. Dual-Branch Parallel Decoding Module (DuaD, the Second Decoding Stage)

After obtaining the decoupled low-frequency global feature $\boldsymbol{g}_{\mathrm{low}}$ and the high-frequency local feature $\boldsymbol{g}_{\mathrm{high}}$, DuaD is designed to process $\boldsymbol{g}_{\mathrm{low}}$ and $\boldsymbol{g}_{\mathrm{high}}$ in parallel.

**Low-frequency branch.** Two MHA blocks are applied to fuse $\boldsymbol{g}_{\mathrm{low}}$ with the agent-level detailed semantic feature $\boldsymbol{E}_{\mathrm{amr}}$ and the scene-level global semantic feature $\boldsymbol{E}_{\mathrm{am}}$, respectively. The fused feature $\boldsymbol{g}_{\mathrm{low}}^{\mathrm{fuse}}$ is then decoded by $\mathrm{MLP}^{\mathrm{tre}}$ to produce the global trend trajectories $\mathbf{Y}_{\mathrm{tre}} \in \mathbb{R}^{N_a \times 2 \times F \times K}$ and the corresponding mode probability scores $\boldsymbol{\Omega} \in \mathbb{R}^{N_a \times K}$. This branch leverages the decoupled future low-frequency features to specifically enhance the modeling of global motion trends.

**High-frequency branch.** Similarly, two MHA blocks fuse the high-frequency local feature $\boldsymbol{g}_{\mathrm{high}}$ with $\boldsymbol{E}_{\mathrm{amr}}$ and $\boldsymbol{E}_{\mathrm{am}}$, yielding $\boldsymbol{g}_{\mathrm{high}}^{\mathrm{fuse}}$. Unlike the low-frequency branch, the high-frequency branch takes both $\boldsymbol{g}_{\mathrm{high}}^{\mathrm{fuse}}$ and the endpoint of the trend trajectory $\boldsymbol{p}_{\mathrm{tre}}^{\mathrm{end}}$ as decoding inputs, so that detail refinement is strictly guided by $\boldsymbol{p}_{\mathrm{tre}}^{\mathrm{end}}$ and does not distort the global trend. The concatenated inputs are decoded by $\mathrm{MLP}^{\mathrm{det}}$ to obtain the local detail trajectories $\mathbf{Y}_{\mathrm{det}} \in \mathbb{R}^{N_a \times 2 \times F \times K}$. This branch uses the decoupled historical high-frequency features to specifically enhance local motion details.

Finally, the global trend and local detail components are combined to obtain the final multimodal predictions:

$$\mathbf{Y}^{\mathrm{final}} = \mathbf{Y}_{\mathrm{tre}} + \mathbf{Y}_{\mathrm{det}}. \quad (10)$$

### 3.6. Time–Frequency Dual-Consistency Loss

All modules are trained jointly in an end-to-end manner with a Winner-Takes-All strategy, where only the mode closest to

the ground-truth trajectory is optimized. Building upon conventional time-domain objectives, frequency-domain losses are additionally introduced. Using the frequency-domain ground truth of future trajectories, stage- and branch-specific loss terms are carefully designed to match the heterogeneous requirements across decoding stages and branches, targeting different frequency components:

$$\mathcal{L}_{\text{total}} = \lambda_1 \mathcal{L}_{\text{pre}} + \lambda_2 \mathcal{L}_{\text{tre}} + \lambda_3 \mathcal{L}_{\text{det}}, \tag{11}$$

where $\mathcal{L}_{\text{pre}}$, $\mathcal{L}_{\text{tre}}$ and $\mathcal{L}_{\text{det}}$ are the loss terms for PreD, the low-frequency branch of DuaD, and the high-frequency branch of DuaD, respectively, and $\lambda_1$, $\lambda_2$ and $\lambda_3$ are the corresponding weighting coefficients. They collaboratively form the Time-Frequency Dual-Consistency Loss.

The time-domain term of $\mathcal{L}_{\text{pre}}$ enforces full-horizon trajectory supervision, while its frequency-domain term primarily constrains the low-frequency components of the trajectory:

$$\mathcal{L}_{\text{pre}} = \lambda_{\text{pre}}^{\text{time}} \Delta_{\text{pre}}^{\text{time}} + \lambda_{\text{pre}}^{\text{low}} \Delta_{\text{pre}}^{\text{fre}}, \tag{12}$$

$$\Delta_{\text{pre}}^{\text{time}} = \left\| \mathbf{Y}_{\text{pre}} - \mathbf{Y}_{\text{gt}} \right\|_1, \tag{13}$$

$$\Delta_{\text{pre}}^{\text{fre}} = \sum_{f \in \mathcal{F}} (1+f)^{-\mu} \left\| \text{DFT}(\mathbf{Y}_{\text{pre}})_f - \text{DFT}(\mathbf{Y}_{\text{gt}})_f \right\|_1, \tag{14}$$

where $\mathbf{Y}_{\text{gt}}$ denotes the ground-truth trajectory, $\mathcal{F}$ is the set of all frequency bins, $\lambda_{\text{pre}}^{\text{time}}$ and $\lambda_{\text{pre}}^{\text{low}}$ are weighting coefficients, and $\| \cdot \|_1$ denotes the Smooth L1 Loss.

$\mathcal{L}_{\text{tre}}$ includes a multimodal classification loss $\mathcal{L}_{\text{cls}}$, a full time-domain trajectory loss $\Delta_{\text{tre}}^{\text{time}}$ and a frequency-domain loss on low-frequency components that reflect global trends $\Delta_{\text{tre}}^{\text{fre}}$. Accordingly, $\mathcal{L}_{\text{tre}}$ is defined as:

$$\mathcal{L}_{\text{tre}} = \lambda_{\text{cls}} \mathcal{L}_{\text{cls}} + \lambda_{\text{tre}}^{\text{time}} \Delta_{\text{tre}}^{\text{time}} + \lambda_{\text{tre}}^{\text{low}} \Delta_{\text{tre}}^{\text{fre}}, \tag{15}$$

$$\Delta_{\text{tre}}^{\text{time}} = \left\| \mathbf{Y}_{\text{tre}} - \mathbf{Y}_{\text{gt}} \right\|_1, \tag{16}$$

$$\Delta_{\text{tre}}^{\text{fre}} = \sum_{f \in \mathcal{F}} (1+f)^{-\mu} \left\| \text{DFT}(\mathbf{Y}_{\text{tre}})_f - \text{DFT}(\mathbf{Y}_{\text{gt}})_f \right\|_1, \tag{17}$$

where $\lambda_{\text{cls}}$, $\lambda_{\text{tre}}^{\text{time}}$ and $\lambda_{\text{tre}}^{\text{low}}$ are weighting coefficients.

For $\mathcal{L}_{\text{det}}$, a residual constraint is introduced in the time-domain term so that the high-frequency branch can stably complement $\mathbf{Y}_{\text{tre}}$ with motion details without altering the global trend. In the frequency-domain term, $\mathcal{L}_{\text{det}}$ primarily enforces consistency between the predicted and ground-truth trajectories in high-frequency components. Accordingly, $\mathcal{L}_{\text{det}}$ is defined as:

$$\mathcal{L}_{\text{det}} = \lambda_{\text{det}}^{\text{time}} \Delta_{\text{det}}^{\text{time}} + \lambda_{\text{det}}^{\text{high}} \Delta_{\text{det}}^{\text{fre}}, \tag{18}$$

$$\Delta_{\text{det}}^{\text{time}} = \left\| \mathbf{Y}_{\text{det}} - (\mathbf{Y}_{\text{gt}} - \mathbf{Y}_{\text{tre}}) \right\|_1, \tag{19}$$

$$\Delta_{\text{det}}^{\text{fre}} = \sum_{f \in \mathcal{F}} (1+f^{\mu}) \left\| \text{DFT}(\mathbf{Y}_{\text{final}})_f - \text{DFT}(\mathbf{Y}_{\text{gt}})_f \right\|_1, \tag{20}$$

where $\lambda_{\text{det}}^{\text{time}}$ and $\lambda_{\text{det}}^{\text{high}}$ are weighting coefficients.

In Eqs. 14, 17, and 20, $\mu > 0$ is a positive scalar that controls the frequency weighting. Here, $f$ denotes the non-negative normalized frequency magnitude of each DFT bin after frequency reordering. Specifically, it makes $\Delta_{\text{pre}}^{\text{fre}}$ and $\Delta_{\text{tre}}^{\text{fre}}$ monotonically decreasing functions of the frequency $f$, while $\Delta_{\text{det}}^{\text{fre}}$ becomes a monotonically increasing function of $f$. This design ensures that $\Delta_{\text{pre}}^{\text{fre}}$, $\Delta_{\text{tre}}^{\text{fre}}$ and $\Delta_{\text{det}}^{\text{fre}}$ specifically target the low-frequency components of the preliminary trajectory, the low-frequency components of the global-trend trajectory, and the high-frequency components of the final trajectory, respectively.

## 4. Experiments and Analysis

### 4.1. Experimental Settings

**Real-World Datasets:** To evaluate the proposed method, experiments are conducted on two widely used large-scale motion forecasting benchmarks, Argoverse 1 and Argoverse 2. Dataset details are provided in Appendix B.1.

**Evaluation Metrics:** Performance is assessed using six standard evaluation metrics: $\text{minADE}_K$, $\text{minFDE}_K$, $\text{MR}_K$, $\text{Brier\_minFDE}_K$, the Brier score $(1-p)^2$, and DAC. $K$ denates the number of trajectory modes in the evaluation metrics, which is set to 6 in this paper. For more details on the evaluation metrics and experimental implementation, refer to Appendices B.2 and B.3, respectively.

### 4.2. Comparison with State-of-the-Art (SOTA)

**Argoverse 1.** Table 1 compares TF-FACE with SOTA methods on the Argoverse 1 benchmark. Overall, TF-FACE achieves a strong balance across accuracy, endpoint stability, mode reliability, probability calibration, and physical feasibility. Specifically, TF-FACE achieves $\text{Brier\_minFDE}_6 = 1.73$, $\text{minADE}_6 = 0.76$, sharing the best performance with HPNet, indicating superior trajectory fitting quality over the full horizon. TF-FACE also achieves $\text{minFDE}_6 = 1.11$, which is comparable to the best HPNet result (1.10) and improves over recent methods such as GoIRL (1.17) and FIM (1.20), reflecting more accurate endpoint prediction.

In terms of multimodal discriminability and reliability, TF-FACE achieves $(1-p)^2 = 0.62$, sharing the best result with GoIRL and outperforming classic baselines such as HiVT (0.67) and LAformer (0.68). This indicates that TF-FACE produces a more concentrated multimodal distribution and assigns higher confidence to the mode closest to the ground truth. TF-FACE further achieves $\text{MR}_6 = 0.107$, sharing the top performance with HPNet, suggesting not only accurate best-mode prediction but also strong average quality across modes. Regarding physical feasibility, TF-FACE achieves the best DAC of 0.9911, indicating that predicted trajectories are largely constrained within drivable areas. This benefit arises from the frequency-domain adaptive encod-

ing module, which aligns scene-level global semantics with local interaction cues via multi-head attention.

*Table 1.* Single-model results on the Argoverse 1 motion forecasting benchmark, ranked by the official metric Brier_minFDE$_6$.

| Model | Brier_min FDE$_6$ ↓ | min FDE$_6$ ↓ | min ADE$_6$ ↓ | $(1-p)^2$ ↓ | DAC ↑ | MR$_6$ ↓ |
|---|---|---|---|---|---|---|
| LaneGCN(2020) | 2.07 | 1.37 | 0.87 | 0.70 | 0.981 | 0.163 |
| LaneRCNN(2021) | 2.14 | 1.45 | 0.90 | 0.69 | 0.990 | 0.123 |
| DenseTNT(2021) | 1.97 | 1.28 | 0.88 | 0.69 | 0.987 | 0.125 |
| SceneTrans(2022) | 1.88 | 1.23 | 0.80 | 0.65 | 0.989 | 0.126 |
| HiVT-128(2022c) | 1.84 | 1.17 | 0.77 | 0.67 | 0.988 | 0.127 |
| Macformer(2023) | 1.83 | 1.21 | 0.82 | 0.62 | 0.986 | 0.126 |
| GANet(2023a) | 1.79 | 1.16 | 0.81 | 0.63 | 0.989 | 0.118 |
| LAformer(2024) | 1.84 | 1.16 | 0.77 | 0.68 | 0.989 | 0.125 |
| SIMPL(2024b) | 1.81 | 1.17 | 0.79 | 0.64 | 0.990 | 0.126 |
| HPNet(2024) | 1.73 | **1.10** | 0.76 | 0.63 | 0.988 | 0.107 |
| FIM(2025a) | 1.83 | 1.20 | 0.82 | 0.65 | 0.990 | 0.125 |
| GoIRL(2025b) | 1.79 | 1.17 | 0.81 | 0.62 | 0.989 | 0.120 |
| **TF-FACE (ours)** | **1.73** | 1.11 | **0.76** | **0.62** | **0.991** | **0.107** |

Without introducing additional training data, a random-seed ensemble is further applied at test time. As shown in Table 2, ensembling consistently improves TF-FACE over its single-model counterpart, and achieves the best Brier_minFDE$_6$ and MR$_6$. Although TF-FACE ranks second to QCNet on minFDE$_6$ and minADE$_6$, TF-FACE significantly outperforms QCNet on the Brier score $(1-p)^2$, indicating that TF-FACE can reduce prediction error while selecting the best trajectory with higher confidence.

*Table 2.* Ensemble-model results on the Argoverse 1 motion forecasting benchmark, ranked by the official metric Brier_minFDE$_6$.

| Model | Brier_min FDE$_6$ ↓ | min FDE$_6$ ↓ | min ADE$_6$ ↓ | MR$_6$ ↓ |
|---|---|---|---|---|
| DenseTNT (2021) | 1.976 | 1.281 | 0.88 | 0.126 |
| Multipath++ (2022) | 1.793 | 1.214 | 0.79 | 0.132 |
| Macformer (2023) | 1.767 | 1.214 | 0.81 | 0.127 |
| SIMPL (2024b) | 1.747 | 1.155 | 0.77 | 0.117 |
| Wayformer (2023) | 1.741 | 1.161 | 0.77 | 0.119 |
| ProphNet (2023b) | 1.694 | 1.134 | 0.76 | 0.110 |
| QCNet (2023) | 1.694 | **1.067** | **0.73** | 0.106 |
| GoIRL (2025b) | 1.695 | 1.126 | 0.78 | 0.110 |
| FIM (2025a) | 1.693 | 1.119 | 0.78 | 0.109 |
| **TF-FACE (ours)** | **1.676** | 1.098 | 0.74 | **0.105** |

**Argoverse 2.** To evaluate the generalizability of TF-FACE, a custom benchmark is constructed by re-splitting the raw clips in Argoverse 2. Specifically, the most recent 20 frames in the observation window are used as input to predict the next 30 frames. Under this setting, TF-FACE is compared with three high-performing models, namely DeMo (Zhang et al., 2024a), QCNet, and FIM. As reported in Table 3, TF-FACE achieves the best performance on the two core metrics Brier_minFDE$_6$ and minFDE$_6$.

TF-FACE is further evaluated under a longer prediction horizon. The model still takes 20 history frames as input, but predicts 60 future frames, and is compared against the

*Table 3.* Single-model results of predicting the next 30 frames on the Argoverse 2 validation set.

| Model | Brier_minFDE$_6$ ↓ | minFDE$_6$ ↓ | $(1-p)^2$ ↓ |
|---|---|---|---|
| QCNet | 1.180 | 0.551 | 0.629 |
| DeMo | 1.169 | 0.543 | 0.626 |
| FIM | 1.131 | 0.530 | **0.626** |
| **TF-FACE (ours)** | **1.092** | **0.485** | 0.607 |

widely used SOTA baseline QCNet on Argoverse 2. To avoid disadvantaging QCNet, the input length is not forced to be identical across methods; instead, QCNet keeps its default 50-frame history input, while TF-FACE uses 20 frames. As shown in Table 4, despite the substantially increased difficulty under the longer horizon, TF-FACE maintains endpoint errors comparable to QCNet and achieves better results on minADE$_6$ and MR$_6$. Given that TF-FACE uses a much shorter history window than QCNet, these results suggest that its long-horizon advantage does not stem from longer observations, but rather from adaptive and controllable utilization of frequency-domain information. We also conducted experiments to test the model's robustness to perturbation, which can be referred to Appendix C.

*Table 4.* Single-model results of predicting the next 60 frames on the Argoverse 2 validation set.

| Model | Brier_min FDE$_6$ ↓ | min FDE$_6$ ↓ | min ADE$_6$ ↓ | MR$_6$ ↓ |
|---|---|---|---|---|
| QCNet | **1.90** | **1.29** | 0.726 | 0.167 |
| **TF-FACE (ours)** | 1.93 | 1.31 | **0.725** | **0.164** |

### 4.3. Ablation Study

To quantify the contribution of each key component in TF-FACE, a series of ablation studies is conducted on the Argoverse 1 validation set. All training and inference settings are kept identical to the full model; only the target component is removed or replaced. Results are reported in Table 5.

Removing the GfreAttn downgrades the adaptive frequency-domain fusion encoder into a time-domain FPN that directly encodes historical trajectories. Without adaptive enhancement of task-relevant frequency bands, the features provided to the decoder become less informative, leading to the most pronounced performance drop. When LFE is ablated, frequency-domain modeling is no longer used to extract future global features from the coarse predictions, and freMLP is replaced with a time-domain MLP, which weakens trend modeling and significantly degrades long-horizon performance. When ablating the high-frequency local feature extraction module, the model can no longer explicitly decouple high-frequency details. Consequently, the high-frequency branch in the dual-branch decoder is removed, and the low-frequency trend trajectory is used as the final prediction. This variant largely preserves the global trend but reduces the capability to capture local details and short-term maneuvers. Finally, removing the DuaD makes

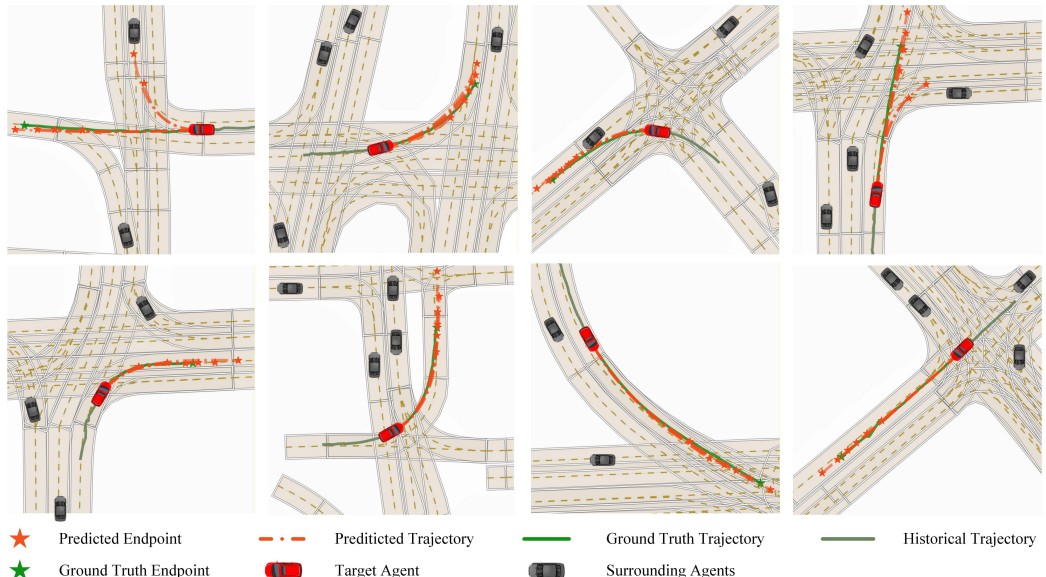

*Figure 4.* Visualizations of TF-FACE on the Argoverse 1 validation set.

the model output the coarse trajectories from Stage I directly, resulting in weaker modeling of both global trends and local details.

*Table 5.* Ablation studies on the Argoverse 1 validation set.

| GfreAttn | LFE | HFE | DuaD | Brier_min FDE$_6$ ↓ | min FDE$_6$ ↓ | MR$_6$ ↓ | min ADE$_6$ ↓ |
|---|---|---|---|---|---|---|---|
| | ✓ | ✓ | ✓ | 1.549 | 0.947 | 0.081 | 0.667 |
| ✓ | | ✓ | ✓ | 1.526 | 0.920 | 0.077 | 0.648 |
| ✓ | ✓ | | ✓ | 1.508 | 0.898 | 0.074 | 0.642 |
| ✓ | ✓ | ✓ | | 1.552 | 0.944 | 0.080 | 0.660 |
| ✓ | ✓ | ✓ | ✓ | **1.496** | **0.890** | **0.071** | **0.635** |

In addition, an ablation study is conducted on the time–frequency dual-consistency loss. Further details of the ablation studies can be found in Appendix D.

### 4.4. Qualitative Analysis

Qualitative visualization is conducted on representative complex traffic scenarios in Argoverse 1, as shown in Figure 4. TF-FACE produces smooth and physically plausible trajectories that closely match the ground truth. Notably, even in narrow road segments or under interference from surrounding agents, the predictions remain well constrained within drivable lanes. Moreover, the predicted modes are tightly concentrated around the ground-truth trajectory. This is attributed to the dedicated low-frequency branch, which enables clear identification and accurate extraction of global trends and behavioral semantics, allowing the model to capture the dominant long-term motion pattern earlier and more stably. The model then applies bounded refinements to local details within the feasible region, yielding a complementary fusion of global trends and local details. Additional qualitative results are provided in Appendix E.

Experiments are also conducted to test inference efficiency, which can be referred to Appendix F.

## 5. Conclusion

This paper proposes TF-FACE, a time-frequency fusion learning framework for multimodal trajectory prediction in autonomous driving. TF-FACE adaptively and controllably exploits frequency-domain information to improve prediction performance, addressing the insufficient modeling of long-term dependencies and short-term motion patterns in purely time-domain learning. To the best of our knowledge, TF-FACE is the first autonomous-driving trajectory prediction model that systematically integrates frequency-domain learning throughout the pipeline, spanning scene encoding, feature extraction, and trajectory decoding. Experiments on large-scale motion forecasting benchmarks demonstrate that TF-FACE achieves state-of-the-art performance, with advantages in overall error, endpoint accuracy, mode selection and distribution, and physical feasibility, while also exhibiting strong robustness to noisy inputs. Moreover, TF-FACE is computationally efficient and meets real-time inference requirements for practical autonomous driving systems.

A current limitation is that the split between low- and high-frequency bands is empirically set based on the distributions of low- and high-frequency data across datasets. Future work will explore learnable band partitioning to improve transferability. In addition, the proposed frequency-domain adaptive and controllable enhancement paradigm will be extended to multimodal information to further improve end-to-end autonomous driving performance.

## Impact Statement

This paper presents work whose goal is to advance the field of machine learning. There are many potential societal consequences of our work, none of which we feel must be specifically highlighted here.

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

# A. Additional Method

## A.1. TF-FACE's Three Types of Input Features

We further process the agents' historical trajectories and map information to derive three types of input features for TF-FACE: time-domain historical trajectories $\text{Traj} \in \mathbb{R}^{N_a \times 2 \times H}$, vectorized map features $\boldsymbol{\mathcal{M}}^{\text{V}} \in \mathbb{R}^{N_m \times B \times d_m}$, and relative spatial poses $\boldsymbol{R} \in \mathbb{R}^{(N_a+N_m) \times (N_a+N_m) \times 5}$ :

(1) The agents' time-domain historical trajectories can be represented as discrete-time sequences of lateral and longitudinal positions; the time-domain historical trajectories of all agents are denoted as $\text{Traj} \in \mathbb{R}^{N_a \times 2 \times H}$.

(2) The vectorized map is used to represent the road topology and geometric structure. Compared with rasterized maps, vectorized maps provide a more compact representation and are computationally more efficient. The vectorized map is denoted as $\boldsymbol{\mathcal{M}}^{\text{V}} \in \mathbb{R}^{N_m \times B \times d_m}$, which is the collection of vectorized representations for all $N_m$ map elements.

(3) To characterize interaction and coupling among all semantic elements in the scene, a relative spatial pose matrix $\boldsymbol{R}$ is constructed over all elements. The relative pose from element $i$ to element $j$, denoted as, $\boldsymbol{r}_{i \rightarrow j}$ is parameterized by the relative displacement vector $\boldsymbol{d}_{i \rightarrow j}$, the heading difference $\alpha_{i \rightarrow j}$, and the relative bearing angle $\beta_{i \rightarrow j}$.

$$\boldsymbol{r}_{\text{i}\rightarrow\text{j}} = [\|\boldsymbol{d}_{i \rightarrow j}\|, \sin(\alpha_{i \rightarrow j}), \cos(\alpha_{i \rightarrow j}), \sin(\beta_{i \rightarrow j}), \cos(\beta_{i \rightarrow j})]. \tag{21}$$

All pairwise relative poses form the relative spatial pose matrix $\boldsymbol{R} \in \mathbb{R}^{(N_a+N_m) \times (N_a+N_m) \times 5}$.

## A.2. Vectorized Map Encoding

Following the point-feature modeling paradigm in PointNet, the vectorized map $\boldsymbol{\mathcal{M}}^{\text{V}}$ containing all $N_m$ map elements is encoded into compact embeddings. Specifically, $\boldsymbol{\mathcal{M}}^{\text{V}}$ is first projected by a linear layer into a higher-dimensional representation $\mathcal{H}^{\text{V}}$. Then, max pooling and nonlinear aggregation are applied to jointly model the geometric patterns and semantic attributes of each map element, yielding a compact vectorized map embedding $\boldsymbol{E}_{\text{map}} \in \mathbb{R}^{N_m \times D}$ :

$$\mathcal{H}^V = \text{MLP}^{\text{map}\,1}\left(\boldsymbol{\mathcal{M}}^{\text{V}}\right), \mathcal{H}^V \in \mathbb{R}^{N_m \times B \times D}, \tag{22}$$

$$\forall \boldsymbol{h}_i \in \boldsymbol{\mathcal{H}}^{\text{V}}, \boldsymbol{h}_i \in \mathbb{R}^{B \times D}, i = \{1, 2, \ldots, N_m\}, \tag{23}$$

$$\overline{\boldsymbol{h}}_i = \boldsymbol{h}_i + \text{MLP}^{\text{map}\,2}\left(\boldsymbol{h}_i \oplus \text{maxpool}\left(\boldsymbol{h}_i(:, D)\right)\right), \overline{\boldsymbol{h}}_i \in \mathbb{R}^{B \times D}, \tag{24}$$

$$\widetilde{\boldsymbol{h}}_i = \text{maxpool}\left(\overline{\boldsymbol{h}}_i(:, D)\right), \widetilde{\boldsymbol{h}}_i \in \mathbb{R}^D, \tag{25}$$

$$\boldsymbol{E}_{\text{map}} = \left[\widetilde{\boldsymbol{h}}_1, \widetilde{\boldsymbol{h}}_2, \ldots, \widetilde{\boldsymbol{h}}_{N_m}\right]. \tag{26}$$

## A.3. Relative Spatial Pose Encoding

An MLP is directly applied to encode the relative spatial pose matrix $\boldsymbol{R}$ over all scene elements, producing the relative-pose embedding $\boldsymbol{E}_{\text{rel}} \in \mathbb{R}^{(N_a+N_m) \times (N_a+N_m) \times D}$:

$$\boldsymbol{E}_{\text{rel}} = \text{MLP}^{\text{rel}}(\boldsymbol{R}). \tag{27}$$

## A.4. Cross-Timescale Integration of Semantic Features in FPN

First, 1D convolutions are used to project the semantic features at each pyramid layer to a unified channel dimension $D$:

$$\overline{\boldsymbol{F}}^{(i)} = \text{Conv1d}_{1 \times 1}^{C_i \rightarrow D}(\boldsymbol{F}^{(i)}), i \in \{1, 2, 3\}, \tag{28}$$

where $C_i$ denotes the original channel dimension of the semantic feature at the $i$-th pyramid layer, and $\overline{\boldsymbol{F}}^{(i)}$ denotes the $i$-th-layer feature after channel projection, with $\overline{\boldsymbol{F}}^{(1)} \in \mathbb{R}^{N_a \times D \times H}$, $\overline{\boldsymbol{F}}^{(2)} \in \mathbb{R}^{N_a \times D \times (H/2)}$ and $\overline{\boldsymbol{F}}^{(3)} \in \mathbb{R}^{N_a \times D \times (H/4)}$. Next, a top-down pathway is adopted: deeper-layer features are upsampled along the temporal dimension via linear interpolation (Upsample) and added to the adjacent shallower-layer features for channel-wise semantic compensation, followed by a 1D residual convolution to aggregate and smooth the fused features:

$$\boldsymbol{U}^{(2)} = \mathrm{Res1d}\left(\overline{\boldsymbol{F}}^{(2)} + \mathrm{Upsample}\left(\overline{\boldsymbol{F}}^{(3)}\right)\right), \tag{29}$$

$$\boldsymbol{U}^{(1)} = \mathrm{Res1d}\left(\overline{\boldsymbol{F}}^{(1)} + \mathrm{Upsample}\left(\boldsymbol{U}^{(2)}\right)\right), \tag{30}$$

where $\boldsymbol{U}^{(2)} \in \mathbb{R}^{N_a \times D \times (H/2)}$ denotes the cross-scale fused feature between deep and middle layer, $\boldsymbol{U}^{(1)} \in \mathbb{R}^{N_a \times D \times H}$ denotes the final cross-scale fused feature aggregated across all layers. $\boldsymbol{U}^{(1)}$ is then further processed by a Res1d block and GfreAttn to obtain the time-frequency fused representation of historical trajectories across multiple temporal scales $\widehat{\boldsymbol{F}} \in \mathbb{R}^{N_a \times D \times H}$ :

$$\widehat{\boldsymbol{F}} = \mathrm{GfreAttn}\left(\mathrm{Res1d}\left(\boldsymbol{U}^{(1)}\right)\right). \tag{31}$$

## B. Additional Experimental Settings

### B.1. Dataset Details

Both Argoverse 1 and Argoverse 2 consist of trajectory sequences collected from real-world urban driving scenarios, providing kinematic states for multiple traffic participants along with HD map context. Argoverse 1 follows the official split with 205,942, 39,472, 78,143 sequences for training, validation, and testing. Each sequence spans 5 s at 10 Hz, using 2 s of observed history to predict 3 s into the future. Argoverse 2 contains approximately 200,000 training and 25,000 validation sequences. Each sequence spans 11 s at 10 Hz, with 5 s of observed history and a 6 s prediction horizon.

### B.2. Evaluation Metrics

Specifically, $\mathrm{minADE}_6$ reports the average Euclidean distance between the optimal predicted trajectory and the ground truth along all time steps among 6 modes, while $\mathrm{minFDE}_6$ measures the average Euclidean distance between the endpoints of the optimal predicted trajectory and the ground truth; $\mathrm{MR}_6$ reports the fraction of samples whose endpoint errors exceed 2 m for all predicted modes; $\mathrm{Brier\_minFDE}_6$ jointly accounts for endpoint error and predicted confidence, assessing the consistency between accuracy and probability; $(1-p)^2$ evaluates how well the predicted probability distribution matches the ground truth; and DAC is the percentage of samples for which at least one predicted mode lies within the drivable area.

### B.3. Implementation Details

The latent dimension is set to $D = 128$, and the number of attention heads is set to 8. The learnable channel-wise gating factor $\gamma$ is initialized to 1.0, and the residual scaling parameter $\delta$ is set to 0.2; The batch size is 64. Training is conducted on Ubuntu 20.04 with an Intel i9-14900KF CPU and 4 NVIDIA RTX 3090 GPUs for efficient large-scale parallel training. The model is trained for 50 epochs using the AdamW optimizer with a piecewise polynomial learning-rate schedule. The learning rate starts at $1\mathrm{e}{-5}$ and is adjusted during training to $5\mathrm{e}{-5}$, $5\mathrm{e}{-4}$, $3\mathrm{e}{-4}$, $1\mathrm{e}{-4}$, and $5\mathrm{e}{-5}$ to balance convergence speed and stability.

## C. Robustness to Perturbation

Under real-world operating conditions, trajectory prediction inevitably suffers from noisy inputs due to upstream perception and localization errors. Therefore, robustness to noisy inputs is a critical capability for trajectory prediction models.

To evaluate robustness, Gaussian noise is injected into the Argoverse 1 validation set. Specifically, the future ground-truth trajectories are kept unchanged, while zero-mean Gaussian noise is added to the historical trajectories in the local coordinate frame. The noise magnitude is controlled by the standard deviation $\sigma$, with $\sigma \in \{0.01, 0.03, 0.05\}$, corresponding to approximately 3–15 cm positional perturbations in real-world settings. This range aligns with typical perception and localization errors in practical autonomous driving systems and covers mild to severe noise conditions.

As shown in Table 6, under mild noise ($\sigma = 0.01$), TF-FACE limits the degradation of all metrics to around 3%. Under moderate noise ($\sigma = 0.03$), the performance drop remains within 8%, substantially better than the purely time-domain baseline LAformer, which exhibits 10%–24% degradation under the same noise level. More notably, under severe noise ($\sigma = 0.05$), TF-FACE degrades by 6.3%, 9.13%, 11.1% on $\mathrm{minADE}_6$, $\mathrm{minFDE}_6$ and $\mathrm{MR}_6$, respectively, whereas LAformer suffers much larger drops of 27.4% / 23.05% / 55.13%. Both TF-FACE and LAformer are trained on the original clean Argoverse 1 training set and are evaluated directly on the noisy validation set without any additional noise-specific training.

TF-FACE's robustness gains are attributed to its incorporation of frequency-domain information. By decoupling low- and high-frequency components, TF-FACE prevents high-frequency local perturbations from corrupting low-frequency global trends, while the GfreAttn further suppresses noise.

*Table 6.* Evaluation of prediction robustness on the noisy validation set of Argoverse 1.

| $\sigma$ | TF-FACE (ours) | | | LAformer | | |
|---|---|---|---|---|---|---|
| | $\mathrm{minADE}_6 \downarrow$ | $\mathrm{minFDE}_6 \downarrow$ | $\mathrm{MR}_6 \downarrow$ | $\mathrm{minADE}_6 \downarrow$ | $\mathrm{minFDE}_6 \downarrow$ | $\mathrm{MR}_6 \downarrow$ |
| 0 | **0.635** | **0.887** | **0.072** | **0.643** | **0.921** | **0.083** |
| 0.01 | 0.649 | 0.916 | 0.074 | 0.65 | 0.929 | 0.086 |
| | $\downarrow 2.20\%$ | $\downarrow 3.27\%$ | $\downarrow 2.77\%$ | $\downarrow 1.03\%$ | $\downarrow 0.94\%$ | $\downarrow 3.14\%$ |
| 0.03 | 0.662 | 0.939 | 0.078 | 0.714 | 1.008 | 0.103 |
| | $\downarrow 4.25\%$ | $\downarrow 5.86\%$ | $\downarrow 8.33\%$ | $\downarrow 10.97\%$ | $\downarrow 9.49\%$ | $\downarrow 24.13\%$ |
| 0.05 | 0.675 | 0.968 | 0.080 | 0.820 | 1.133 | 0.129 |
| | $\downarrow 6.30\%$ | $\downarrow 9.13\%$ | $\downarrow 11.10\%$ | $\downarrow 27.40\%$ | $\downarrow 23.05\%$ | $\downarrow 55.13\%$ |

# D. Additional Ablation Study

## D.1. Ablation Study of the Time–Frequency Dual-Consistency Loss

Table 7 reports the impact of applying frequency-domain dual constraints at different decoding stages while keeping the model architecture unchanged. $\Delta^{\mathrm{time}}$ denotes the time-domain constraints applied across all decoding stages. Using only $\Delta^{\mathrm{time}}$ still yields reasonable performance. When the low-frequency constraints $\Delta^{\mathrm{fre}}_{\mathrm{pre}}$ and $\Delta^{\mathrm{fre}}_{\mathrm{tre}}$ are removed, the model's ability to capture long-range dependencies is weakened, leading to a more noticeable increase in $\mathrm{minADE}_6$, $\mathrm{Brier\_minFDE}_6$ and $\mathrm{minFDE}_6$. When the high-frequency constraint $\Delta^{\mathrm{fre}}_{\mathrm{det}}$ is removed, the model is less capable of refining local motion details, and the accumulation of local errors leads to degraded performance. Notably, while enforcing high-frequency supervision improves fine-grained details, it can introduce slight endpoint perturbations for a small subset of samples. As a result, the variant without $\Delta^{\mathrm{fre}}_{\mathrm{det}}$ may occasionally achieve a lower $\mathrm{minFDE}_6$ than the full loss.

*Table 7.* Ablation studies of the time-frequency dual-consistency loss on the Argoverse 1 validation set.

| $\Delta^{\mathrm{time}}$ | $\Delta^{\mathrm{fre}}_{\mathrm{pre}}$ | $\Delta^{\mathrm{fre}}_{\mathrm{tre}}$ | $\Delta^{\mathrm{fre}}_{\mathrm{det}}$ | $\mathrm{Brier\_minFDE}_6 \downarrow$ | $\mathrm{minFDE}_6 \downarrow$ | $\mathrm{minADE}_6 \downarrow$ |
|---|---|---|---|---|---|---|
| ✓ | | | | 1.515 | 0.913 | 0.645 |
| ✓ | | ✓ | ✓ | 1.503 | 0.893 | 0.637 |
| ✓ | ✓ | | ✓ | 1.508 | 0.897 | 0.639 |
| ✓ | ✓ | ✓ | | 1.501 | **0.886** | 0.639 |
| ✓ | ✓ | ✓ | ✓ | **1.496** | 0.890 | **0.635** |

## D.2. Visualization of the Ablation Results for the GfreAttn

In the ablated model without the GfreAttn, FAE degenerates into a time-domain FPN that directly encodes the agents' time-domain historical trajectories. As shown in Figure 5, this variant exhibits clear degradation in both global trend estimation and local detail capture compared to the full TF-FACE model. The reason is that, although frequency-domain information is introduced during decoding via the DFT, frequency-domain learning is not incorporated at the encoding stage; consequently, the decoding still relies on representations that are essentially time-domain features. Similar to prior frequency-domain attempts in autonomous driving, this ablated variant does not achieve a deep and systematic frequency-domain learning pipeline, but remains at the level of directly using raw frequency-domain signals. In contrast, with the learnable GfreAttn, TF-FACE can adaptively enhance or suppress different frequency bands and jointly represent long-term dependencies and transient motion patterns, validating the effectiveness of the proposed GfreAttn.

## D.3. Visualization of the Ablation Results for LFE

In the ablated variant of LFE, frequency-domain modeling is no longer used when extracting future low-frequency global features from the coarse predicted trajectories, and freMLP is replaced with a standard time-domain MLP. In the left-turn-at-intersection scenario shown in Figure 6, the ablated model aligns reasonably well with the ground-truth trajectory at early

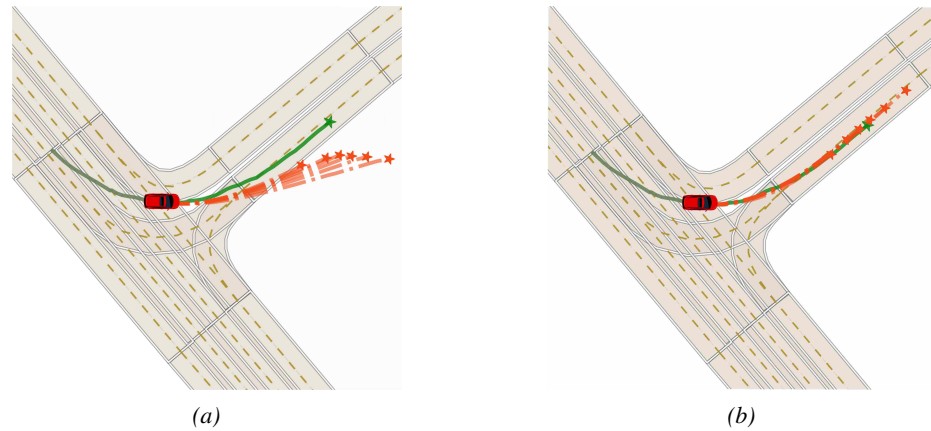

*Figure 5.* Qualitative examples for the GfreAttn ablation. (a) Model without GfreAttn. (b) Full TF-FACE.

prediction steps; however, as the horizon extends, errors accumulate and lead to noticeable deviations in the long-term trend and endpoint. This indicates that a conventional time-domain MLP is less effective at capturing future low-frequency trends.

In contrast, the full TF-FACE model leverages freMLP to more accurately capture future low-frequency global features in the frequency-domain, thereby maintaining correct motion trends over long horizons. These results validate the effectiveness of the proposed LFE.

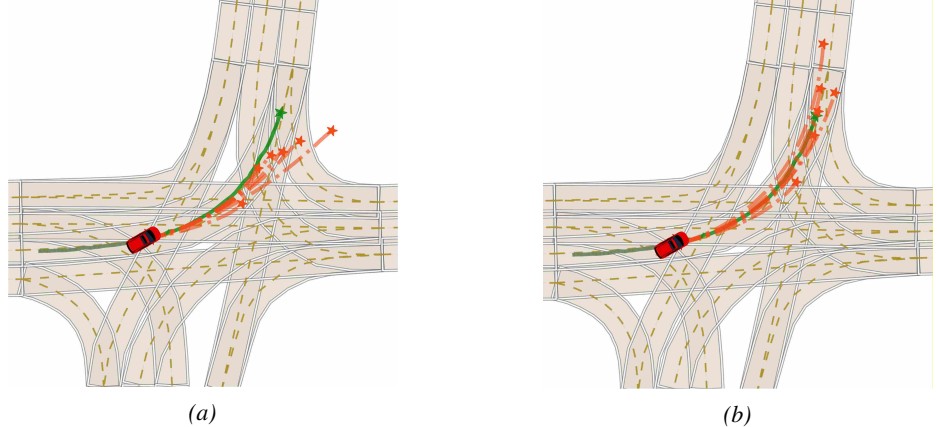

*Figure 6.* Qualitative examples for the LFE ablation. (a) Model without the LFE. (b) Full TF-FACE.

### D.4. Visualization of the Ablation Results for HFE

In the ablated variant of HFE, historical high-frequency local features are unavailable. Consequently, the high-frequency branch in DuaD is removed, and the global trend trajectory produced by the low-frequency branch is used as the final prediction. As shown in Figure 7, because low-frequency global features are still extracted by LFE and utilized by the low-frequency branch, the ablated model remains consistent with the ground truth in terms of the overall motion trend. However, without detail refinement driven by high-frequency features, the model loses fine-grained behavior in turning segments and regions with large local curvature changes, manifested as inaccurate turning timing, overly smoothed curvature variations, and accumulated local errors.

In contrast, by jointly leveraging low-frequency global features and high-frequency local features, the full TF-FACE model maintains accurate global trends while precisely capturing detailed trajectory shapes and dynamic variations. These results validate the effectiveness of the proposed HFE.

## D.5. Visualization of the Ablation Results for DuaD

In the ablated variant of the DuaD, the preliminary trajectories from the first stage are directly used as the final predictions. In the complex intersection scenario shown in Figure 8, the ablated model lacks targeted modeling of future low-frequency global features and therefore fails to capture the agent's global motion trend. This leads to mode divergence around the intersection region and noticeable deviation from the ground truth. Meanwhile, without leveraging high-frequency features, the model cannot capture local motion-detail variations, and local offsets gradually accumulate over the prediction horizon.

In contrast, benefiting from controllable decoupling and coupling of low- and high-frequency features, the full TF-FACE model produces multimodal predictions that are clearly concentrated on two behavior semantics in the same scene, namely going straight and turning right. Four modes accurately capture the global trend, and TF-FACE also recovers fine-grained trajectory details at key locations where local curvature changes due to turning. These results validate the effectiveness of the proposed DuaD.

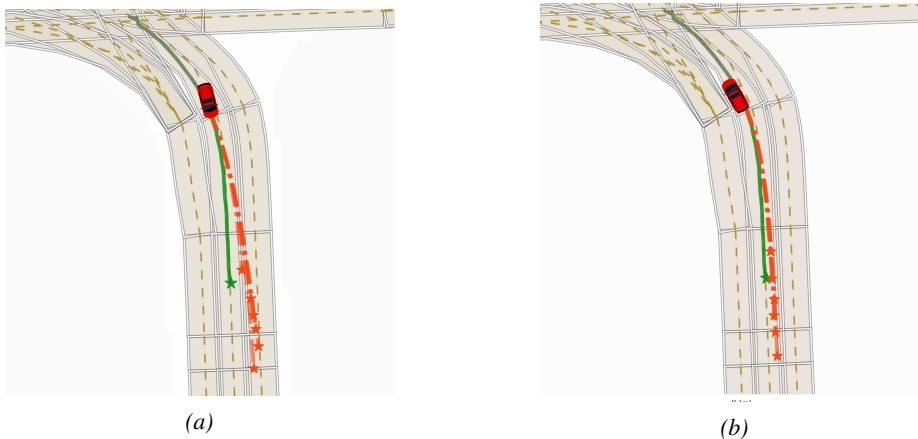

*(a)*            *(b)*

*Figure 7.* Qualitative examples for the ablation of HFE. (a) Model without HFE. (b) Full TF-FACE.

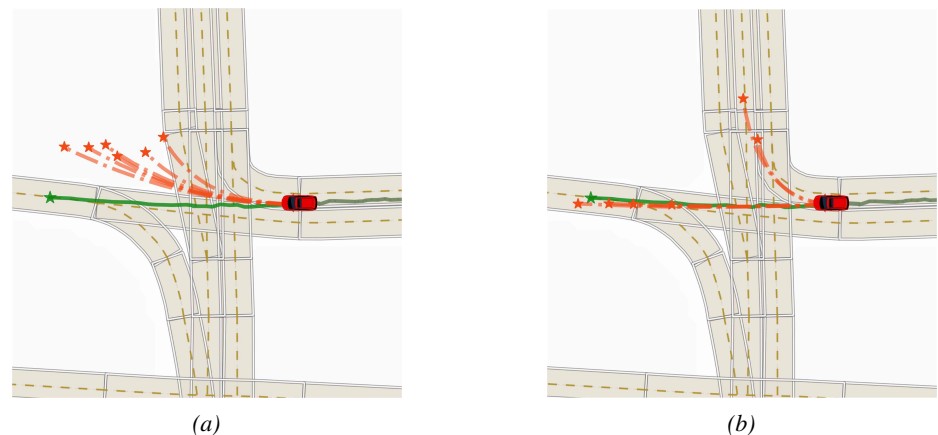

*(a)*            *(b)*

*Figure 8.* Qualitative examples for the ablation of DuaD. (a) Model without DuaD. (b) Full TF-FACE.

## D.6. Analysis of Key Model Hyperparameters

### D.6.1. Analysis of Loss Weights for the Multi-Stage Objective:

We study the loss weights in the multi-stage objective in Eq. 11. Specifically, $\lambda_1$, $\lambda_2$ and $\lambda_3$ weight the PreD loss $\mathcal{L}_{\mathrm{pre}}$, the low-frequency branch loss $\mathcal{L}_{\mathrm{tre}}$ and the high-frequency branch loss $\mathcal{L}_{\mathrm{det}}$ in DuaD, respectively. Table 8 reports results on the Argoverse 1 validation set under different weight settings. Reducing $\lambda_1$ while increasing $\lambda_2$ and $\lambda_3$ consistently improves performance, suggesting that second-stage trend modeling and detail refinement are more critical than tightly supervising the preliminary trajectory. In particular, increasing $\lambda_2$ yields more pronounced gains in endpoint accuracy, while a moderate $\lambda_3$ further reduces average error and improves the quality of the multimodal probability distribution. Overweighting $\lambda_3$

slightly degrades performance, indicating the need to balance trend and detail losses. Based on these observations, we set $\lambda_1 = 0.2, \lambda_2 = 0.6, \lambda_3 = 0.2$ throughout the paper.

*Table 8.* Multi-stage loss weight analysis on the Argoverse 1 validation set.

| $\lambda_1 (\mathcal{L}_{\mathrm{pre}})$ | $\lambda_2 (\mathcal{L}_{\mathrm{tre}})$ | $\lambda_3 (\mathcal{L}_{\mathrm{det}})$ | Brier minFDE$_6$ $\downarrow$ | minFDE$_6$ $\downarrow$ | minADE$_6$ $\downarrow$ |
|---|---|---|---|---|---|
| 0.3 | 0.6 | 0.1 | 1.508 | 0.903 | 0.643 |
| 0.3 | 0.5 | 0.2 | 1.503 | 0.900 | 0.640 |
| 0.3 | 0.4 | 0.3 | 1.510 | 0.906 | 0.640 |
| 0.2 | 0.7 | 0.1 | 1.499 | 0.894 | 0.638 |
| 0.2 | 0.6 | 0.2 | 1.496 | 0.890 | 0.635 |
| 0.2 | 0.5 | 0.3 | 1.502 | 0.898 | 0.636 |

### D.6.2. Analysis of initialization of the channel-wise gating factor:

We analyze the initialization of the channel-wise gating factor $\gamma$ in GfreAttn. Although $\gamma$ is learnable, its initial value controls the strength of high-frequency injection at the beginning of training and can influence early optimization dynamics. As shown in Table 9, TF-FACE is not highly sensitive to its initialization, and initializing $\gamma$ to 1.0 yields the best performance. In contrast, overly large initial values may excessively amplify high-frequency perturbations before the gating mechanism is fully trained, leading to a slight performance drop. Based on this study, we set the initial value of $\gamma$ to 1.0.

*Table 9.* Analysis of the initialization of the frequency-domain gating factor on the Argoverse 1 validation set.

| Channel-wise gating factor initial value $\gamma$ | Brier minFDE$_6$ $\downarrow$ | minFDE$_6$ $\downarrow$ | minADE$_6$ $\downarrow$ |
|---|---|---|---|
| 0.5 | 1.501 | 0.894 | 0.638 |
| 1.0 | 1.496 | 0.890 | 0.635 |
| 1.5 | 1.505 | 0.899 | 0.639 |

### D.6.3 Analysis of the frequency-domain threshold in the high-pass filter:

We investigate the frequency-domain threshold of high-pass filter, i.e., the split between low- and high-frequency bands, which determines which frequency components are used for trajectory detail refinement. As shown in Table 10, performance is also not highly sensitive to this threshold, and the best results are achieved with a 4:6 low-/high-frequency split. If the low:high ratio is too large, only very high-frequency components are retained, which may overly filter out maneuver-related information and lead to insufficient detail representation. Conversely, if the low:high ratio is too small, more relatively low-frequency components are preserved, which can introduce redundancy and slightly degrade performance. Based on this analysis, we set the high-pass filter threshold to 4:6.

*Table 10.* Analysis of the high-pass filter frequency threshold on the Argoverse 1 validation set.

| High-pass filter frequency threshold (low:high) | Brier minFDE$_6$ $\downarrow$ | minFDE$_6$ $\downarrow$ | minADE$_6$ $\downarrow$ |
|---|---|---|---|
| 5:5 | 1.499 | 0.892 | 0.637 |
| 4:6 | 1.496 | 0.890 | 0.635 |
| 3:7 | 1.504 | 0.896 | 0.640 |

## E. Additional Qualitative Analysis

### E.1. Qualitative Analysis of the GfreAttn

To examine the interpretability of TF-FACE's frequency-domain modeling, we visualize the internal GfreAttn. Figure 9 shows the distribution of the channel-wise gating factor $\gamma$ across the three FPN layers. As the network goes deeper, the distribution of $\gamma$ becomes sparser and exhibits more pronounced channel-wise amplification or suppression. This provides an intuitive view of how TF-FACE, within the FPN hierarchy, progressively and adaptively extracts band-specific features from shallow to deep layers via GfreAttn.

Furthermore, Figure 10 compares the amplitude-frequency spectra before and after high-frequency gating in GfreAttn, i.e., before and after modulating the high-frequency component $\mathbf{V}_{T,Attn}^{\mathrm{fre,high}}$ with the channel-wise gating factor $\gamma$. Before applying high-frequency gating, spectral energy is largely concentrated in the low-frequency region (blue curve), which is

consistent with the fact that low-frequency components dominate the overall motion of trajectory sequences. This indicates that GfreAttn appropriately assigns higher weights to low-frequency components. However, as the FPN is stacked layer by layer, high-frequency components can undergo systematic attenuation due to convolutional smoothing. To counteract this effect, TF-FACE introduces the learnable channel-wise gating factor $\gamma$ to modulate the amplitude of high-frequency components, thereby maintaining sustained attention to high-frequency signals for subsequent motion-detail refinement. As shown in Figure 10, after high-frequency gating, the amplitude of high-frequency components increases substantially (red curve), and spectral energy shifts toward higher frequencies. The quantitative gain brought by high-frequency gating is shown by the green curve.

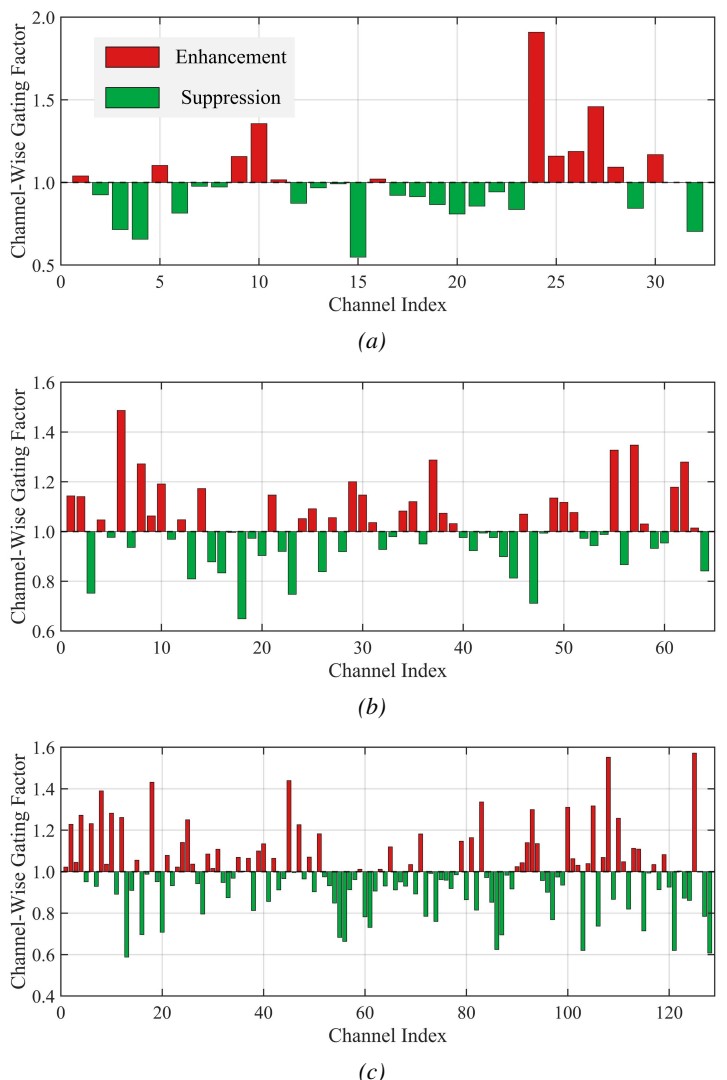

*Figure 9.* Visualization of GfreAttn across FPN layers. (a) Shallow layer. (b) Middle layer. (c) Deep layer.

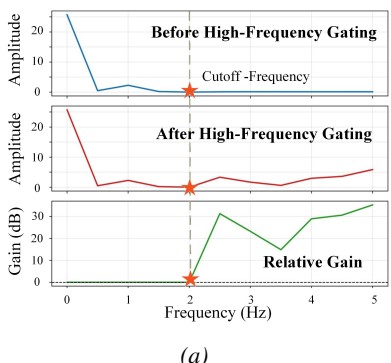
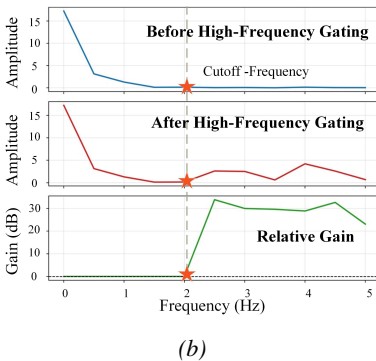

*(a)*                                     *(b)*

*Figure 10.* Visualization of the high-frequency gating effect in the GfreAttn. (a) Example 1. (b) Example 2.

### E.2. Qualitative Analysis of the Two-Stage Decoding Architecture

To analyze how the two-stage decoding architecture behaves during trajectory prediction, we visualize the time-domain and frequency-domain characteristics of the Stage-I preliminary trajectory $\mathbf{Y}_{\mathrm{pre}}$, the Stage-II low-frequency trend $\mathbf{Y}_{\mathrm{tre}}$, the final trajectories obtained by composing the low- and high-frequency branches in Stage II, and the ground-truth trajectories $\mathbf{Y}_{\mathrm{gt}}$, as shown in Figure 11.

In the time-domain, across different behavior semantics, the preliminary trajectory $\mathbf{Y}_{\mathrm{pre}}$ captures the rough future motion direction but often exhibits noticeable global shifts or inaccurate curvature. Compared with $\mathbf{Y}_{\mathrm{pre}}$, the trend trajectories $\mathbf{Y}_{\mathrm{tre}}$, decoded from reliable future low-frequency features, are closer to $\mathbf{Y}_{\mathrm{gt}}$ in both overall shape and endpoint location. The final predictions $\mathbf{Y}_{\mathrm{final}}$ further refine motion details using high-frequency local features, yielding trajectories that are the most similar to $\mathbf{Y}_{\mathrm{gt}}$ in all examples.

In the frequency-domain, the error spectrum of $\mathbf{Y}_{\mathrm{pre}}$ shows the most prominent energy peaks in the low-frequency region, indicating that its dominant errors stem from trend-level deviations. The trend trajectory $\mathbf{Y}_{\mathrm{tre}}$ substantially reduces error magnitudes in low-frequency bands, suggesting that the low-frequency branch effectively captures and reconstructs global motion trends and provides more reliable trend guidance. The final trajectory $\mathbf{Y}_{\mathrm{final}}$ achieves the lowest error magnitude across all frequency bands, showing that incorporating high-frequency local features further compensates for residual local errors that remain after low-frequency trend modeling.

Overall, Stage I, the Stage-II low-frequency branch, and the Stage-II high-frequency branch play complementary roles with clear division of labor. Their collaboration enables TF-FACE to balance global plausibility and local fidelity, providing qualitative evidence for the rationale and interpretability of the proposed two-stage decoding pipeline.

## F. Inference Efficiency and Real-Time Performance

To assess the feasibility of TF-FACE in practical autonomous driving systems, inference efficiency is evaluated. With batch size 1 and float32 precision, TF-FACE achieves an average end-to-end latency of approximately 37 ms (about 26 Hz), which is well above the minimum real-time requirement for trajectory prediction ($\geq 10\,\mathrm{Hz}$). This indicates that TF-FACE can be deployed in real-world autonomous driving systems without additional efficiency optimizations. Furthermore, TF-FACE is compared with state-of-the-art methods on the Argoverse 1 validation set in terms of the accuracy–efficiency trade-off. As shown in Figure 12, TF-FACE achieves a favorable balance, delivering the highest prediction accuracy while maintaining high inference efficiency.

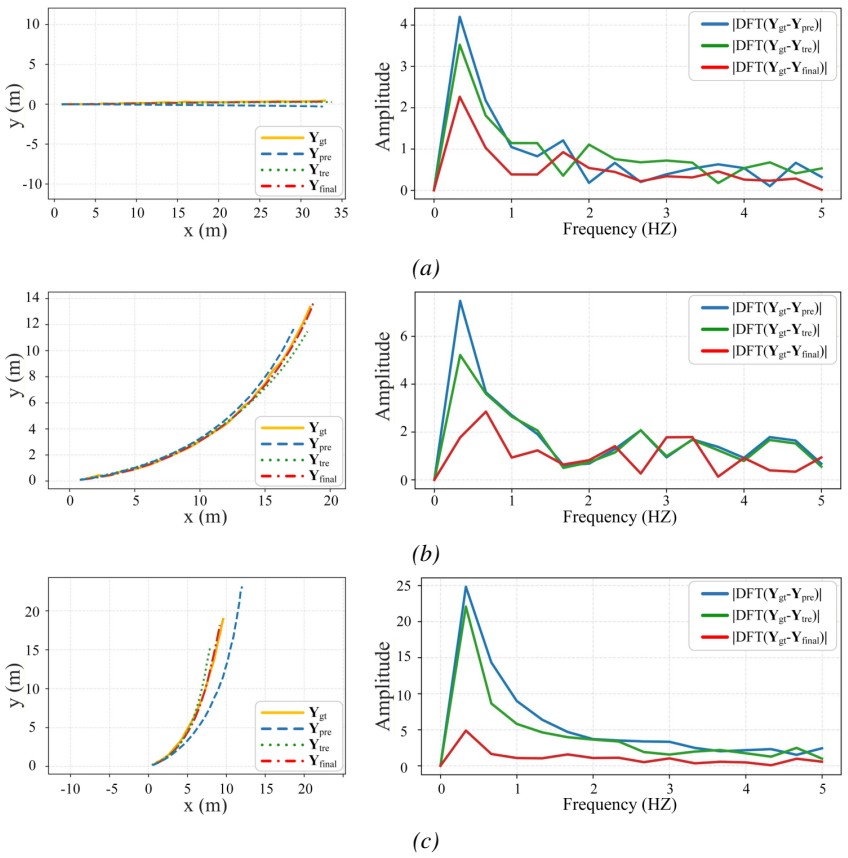

*Figure 11.* Visualization of the two-stage decoding pipeline. (a) Example 1 (time-domain). Example 1 (frequency-domain). (b) Example 2 (time-domain). Example 2 (frequency-domain). (c) Example 3 (time-domain). Example 3 (frequency-domain).

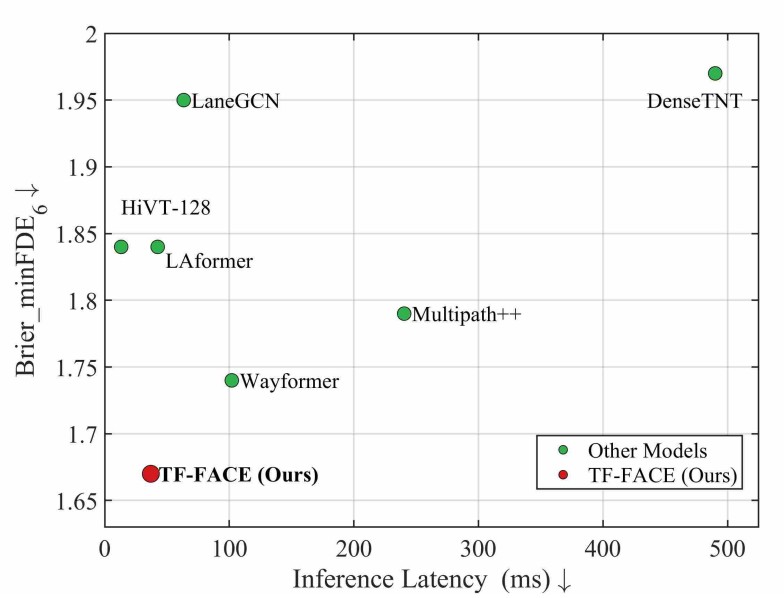

*Figure 12.* Inference efficiency and real-time performance on Argoverse 1.

