# OpenReview forum: "TF-FACE: Time-Frequency Fusion Learning via Frequency-Domain Adaptive and Controllable Enhancement for Trajectory Prediction"
_ICML.cc/2026/Conference — ICML 2026 regular_

### Official Review · Reviewer_b5sW · 2026-03-09

**Soundness:** 3
**Presentation:** 3
**Significance:** 3
**Originality:** 3
**Overall Recommendation:** 4
**Confidence:** 3

**Summary:**

TF-FACE is a time-frequency fusion learning framework designed for multimodal trajectory prediction in autonomous driving. The architecture incorporates a Frequency-domain Adaptive Fusion Encoding (FAE) module with gated frequency-domain attention to extract features across different temporal scales . It utilizes a two-stage decoding pipeline where a preliminary trajectory is first generated and then refined through decoupled low-frequency global trend and high-frequency local detail branches . The model is optimized using a time-frequency dual-consistency loss that matches heterogeneous requirements across different decoding stages and branches.

**Compliance With Llm Reviewing Policy:**

Affirmed.

**Key Questions For Authors:**

Please refer to the section above on weaknesses.

**Limitations:**

yes

**Strengths And Weaknesses:**

Strengths:

1) The framework represents a comprehensive approach by integrating frequency-domain learning throughout scene encoding, feature extraction, and trajectory decoding.

2) The model achieves state-of-the-art accuracy on benchmarks like Argoverse 1 and 2 while maintaining an average end-to-end inference latency of approximately 37 ms, which satisfies real-time requirements.

Weaknesses:

1) The two-stage decoding pipeline involves a significant cascading dependency risk, where the trend refinement in Stage II relies heavily on the quality of the preliminary trajectory $Y_{pre}$ produced in Stage I. Qualitative visualizations indicate that $Y_{pre}$ often exhibits global shifts or inaccurate curvatures. If Stage I produces significant deviations in low-frequency components, the subsequent Low-Frequency Global Trend Feature Extraction (LFE) module might reinforce these erroneous signals, leading to final predictions that diverge from the ground truth. Could the authors provide a formal stability analysis or sensitivity experiments regarding $Y_{pre}$ deviations? When $Y_{pre}$ is subjected to manually injected perturbations, does the model possess self-correction capabilities?

2) The ratio for splitting low and high frequency bands is currently a fixed empirical value based on specific dataset distributions. Different traffic environments, such as highways versus busy intersections, exhibit vastly different motion frequency characteristics. A fixed ratio may fail to effectively decouple non-periodic or sudden maneuvers. How does the model ensure the universality of this partitioning ratio during the inference phase when ground truth labels are unavailable? Could the authors discuss whether the partition boundary should be dynamically adjusted for agents with different physical dynamics, such as fast-moving vehicles compared to slow pedestrians?

3) The framework introduces numerous complex loss function coefficients ($\lambda_1, \lambda_2, \lambda_3$) and a frequency weight scalar $\mu$. Experiments show that model performance is highly sensitive to these hyperparameters. The absence of an automated parameter tuning mechanism significantly increases the practical cost of deployment. As the number of agents $N_a$ in a scene increases, do the frequent DFT/IDFT transformations and multi-head attention calculations lead to a non-linear growth in computational overhead? Please provide inference latency curves across different scales of agent populations.

4) According to Table 7 in Appendix E.1, the results show an insignificant performance gap between the pre-ablation and post-ablation experiments. This raises concerns about whether the trade-off between the methodology overhead and the actual gain is reasonable.

---

> ### Author Rebuttal · Authors · 2026-03-31
>
> We sincerely thank the reviewer for the careful and constructive comments. We address each concern below.
>
> 1. Q1
>
> We agree that Stage II decoding depends on the preliminary trajectory Ypre produced in Stage I. Actually, Stage I is already trained under direct supervision of ground truth and can itself produce reasonably accurate predictions. In many baselines, the output of this stage is already treated as the final output. However, in our framework, the output of Stage I (Ypre) is further used by Stage II.
>
> Specifically, in Stage II, the low-frequency feature is extracted from Ypre and is further fused with other scene-level and interaction-aware features, and the final output is also constrained by dedicated loss terms. Therefore, Stage II is designed to correct and complement the Stage I prediction, rather than mechanically amplifying its errors.
>
> Regarding manually perturbing Ypre, we believe this is not the most realistic stability test, because Ypre is an internal intermediate representation rather than an input variable. In trajectory prediction, the realistic sources of perturbation arise from original observations. Therefore, we conducted experiments with perturbation there, and Table 6 shows that TF-FACE remains robust under noisy inputs.
>
> 2. Q2
>
> We supplemented additional experiments by varying the low-/high-frequency split from 1:9 to 9:1:
>
> Brier_minFDE6: 1.519, 1.507, 1.504, 1.496, 1.499, 1.505, 1.515, 1.520, 1.523
>
> minFDE6: 0.908, 0.899, 0.896, 0.890, 0.892, 0.898, 0.906, 0.908, 0.910
>
> The results remain strong within a relatively coarse range, indicating that the model is not highly sensitive to the exact partition boundary. We believe the robustness mainly comes from two aspects of our design: (1) GfreAttn adaptively reweights different frequency components; (2) the low-/high-frequency branches are coupled again in a complementary manner in Stage II, reducing reliance on a finely tuned manual split.
>
> We have also explored adaptive frequency partition modules with dynamically learned boundaries. However, lightweight designs bring little benefit, while stronger variants increase the parameter count from 6.1 to 7.5M (+23%) and the average latency from 37 to 44 ms (+18%), with only about 1% improvement. Therefore, the current design provides a better practical trade-off between robustness and efficiency.
>
> We recognize the reviewer’s concern about different agent dynamics. Since TF-FACE is trained on Argoverse 1/2 with diverse agent types and interaction patterns, it can handle such dynamic variation to a certain extent.
>
> 3. Q3
>
> We appreciate this important concern. The loss design is not heuristic, but follows the role of each stage in the framework. Since the final goal is to obtain the refined trajectory, the Stage II losses should receive larger weights than the Stage I preliminary loss, i.e., λ2+λ3>>λ1. Moreover, since high-frequency branch refines details on top of trend modeling, we set λ2>λ3. Under this principle, only limited tuning is needed. Empirically, the model is relatively stable within a reasonable range of these settings, but degrades noticeably when the weights violate the above design principle. For example, (0.5,0.3,0.2) gives 0.915/0.645(minFDE6/minADE6), while (0.2,0.3,0.5) degrades to 0.923/0.647. As shown in Table 8, even the weaker settings remain competitive with, or stronger than, many strong baselines. This suggests that the main performance gain comes from the overall architecture, while the hyperparameters mainly control the balance among its components. As for μ, it is introduced to explicitly control the relative emphasis on low-/high-frequency bands in the frequency-domain loss. Regarding the value, the current setting is a common practice of a general moderate magnitude.
>
> We further analyzed the latency on the Argoverse 1 validation set across different agent population ranges. Specifically, there are 16,525/15,399/6,420/1,003/53 scenes with 1-10/11-20/21-30/31-40/41-50 agents, and the corresponding average inference latency is 36.8/37.2/38.0/39.8/42.5 ms, respectively. These show that although runtime increases with scene complexity, the growth remains moderate, and TF-FACE still maintains practical real-time efficiency.
>
> 4. Q4
>
> We would like to clarify that Table 7 is designed to evaluate the effect of the auxiliary time-frequency loss terms, rather than the full contribution of the overall architecture. Therefore, the results there should be interpreted as the gain brought by improved supervision, not as the entire gain of TF-FACE. From this perspective, we believe the gain is non-trivial: compared with the pure time-domain loss, adding only the frequency-domain loss terms already brings a clear and consistent improvement on this dataset, without any increase in model size or test-time complexity.
>
> If accepted, we will make the code publicly available and add all necessary supplementary materials in the camera-ready version to ensure full reproducibility.

---

### Official Review · Reviewer_P8Gi · 2026-03-12

**Soundness:** 3
**Presentation:** 3
**Significance:** 2
**Originality:** 3
**Overall Recommendation:** 4
**Confidence:** 3

**Summary:**

This paper proposes a **time–frequency fusion learning framework** called **TF-FACE** for autonomous driving trajectory prediction. The authors point out that existing **pure time-domain methods** often struggle to simultaneously model **long-term dependencies (low-frequency trends)** and **short-term dynamics (high-frequency details)**.

To address this issue, TF-FACE systematically integrates **frequency-domain learning** into the **end-to-end trajectory prediction pipeline**. The framework employs a **Gated Frequency Attention mechanism (GfreAttn)** combined with a **Feature Pyramid Network (FPN)** to adaptively enhance or suppress different frequency bands, thereby alleviating the **spectral bias** commonly observed in deep learning models.

The model operates in two stages. The **first stage** generates preliminary trajectories. The **second stage** decouples the trajectory into **global trends and local details** through two modules: **Low-Frequency global trend Extraction (LFE)** and **High-Frequency local feature Extraction (HFE)**. These two branches process the trajectory in parallel, enabling the model to learn accurate representations in both the **time domain and frequency domain**.

Experimental results show that **TF-FACE achieves state-of-the-art (SOTA) performance** on both **Argoverse 1** and **Argoverse 2** datasets. The model performs strongly on prediction error metrics and **confidence estimation (Brier Score)**, while also maintaining **real-time inference capability**.

**Compliance With Llm Reviewing Policy:**

Affirmed.

**Key Questions For Authors:**

1. **Sensitivity to cut-off frequency:**
   How sensitive is the model performance to the **cut-off frequency used to separate low- and high-frequency components**? Have experiments been conducted to compare different threshold settings?

2. **Preserving meaningful high-frequency signals:**
   While **GfreAttn suppresses high-frequency noise**, how does it ensure that it does **not suppress critical high-frequency signals** that represent real motion changes, such as **sudden braking or emergency maneuvers**?

3. **Performance with short history trajectories:**
   When the **historical observation window is very short** (e.g., when an agent has just entered the field of view), does the **stability of frequency-domain transformations** degrade, and how does this affect **prediction accuracy**?

**Strengths And Weaknesses:**

## Strengths

- This work is the **first trajectory prediction model that systematically integrates frequency-domain learning throughout the entire pipeline**, including encoding, feature extraction, and decoding, demonstrating strong novelty.
- The proposed **GfreAttn mechanism** effectively addresses the issue that deep neural networks tend to **under-attend to high-frequency information**, while the **dual-branch decoder** provides an intuitive physical decoupling of trajectory motion into **global trends and local variations**.
- The method achieves **leading performance on the Argoverse benchmark**, particularly with significant improvements in **Brier Score**, indicating that the model not only predicts more accurately but also selects the most plausible trajectory with higher confidence.
- The paper includes **extensive validation experiments** on loss weights, initialization parameters, and the effectiveness of individual modules.

---

## Weaknesses

- The **threshold separating low-frequency and high-frequency components** is determined empirically based on dataset statistics. The lack of an **adaptive or learnable mechanism** may limit the model’s **generalization ability across different scenarios**.
- Compared with pure time-domain models such as **MLP** or **Transformer-based architectures**, TF-FACE introduces **multiple DFT/IDFT transformations and decoupled branches**, which increases the **implementation and debugging complexity**.
- Although frequency decoupling works well in typical scenarios, in **non-typical cases involving very abrupt motion over extremely short time intervals**, high-frequency details may become highly entangled with global trends. The paper provides limited discussion on such **extreme edge cases**.

---

> ### Author Rebuttal · Authors · 2026-03-31
>
> We sincerely thank the reviewer for the careful, insightful, and constructive comments. We have carefully checked the issues raised and will revise the manuscript accordingly. We address each concern below.
>
> 1. Q1
>
> We have included preliminary comparisons with different threshold settings in the appendix, and we further supplement the results here. Specifically, when varying the low/high-frequency split from 1:9 to 9:1, we obtain:
>
> Brier_minFDE6: 1.519, 1.507, 1.504, 1.496, 1.499, 1.505, 1.515, 1.520, 1.523
>
> minFDE6: 0.908, 0.899, 0.896, 0.890, 0.892, 0.898, 0.906, 0.908, 0.910
>
> The performance remains stable across a broad range of partition ratios, indicating low sensitivity to the exact cut-off. This robustness is largely due to GfreAttn, which adaptively reweights frequency components and reduces reliance on fine-grained partition tuning.
>
> We also note that this limitation has already been acknowledged in the conclusion. To further examine this issue, we implemented adaptive frequency partition modules with dynamically learned boundaries. However, lightweight designs bring negligible gains, while stronger variants increase the parameter count from 6.1M to 7.5M (+23%) and the average latency from 37 ms to 44 ms (+18%), with only about 1% improvement. Therefore, although we have explored this direction, the overall trade-off is unfavorable in practice. We retain the current design for a better balance between robustness and efficiency.
>
> 2. Q2
>
> We would like to clarify that GfreAttn does not simply suppress high-frequency components as noise. Instead, we specifically design this frequency-domain attention mechanism to adaptively assign different weights to different frequency components during training. In this way, frequency components that are consistently informative for prediction can be preserved or emphasized, while irrelevant noisy components are downweighted. Therefore, meaningful high-frequency signals corresponding to real motion changes are not removed by design, but are adaptively retained when they are beneficial for prediction.
>
> This is also supported by our ablation results. Without GfreAttn, the performance drops to Brier_minFDE6/minFDE6/minADE6 = 1.549/0.947/0.667, while with GfreAttn the corresponding results improve to 1.496 / 0.890 / 0.635. This shows that GfreAttn is a fundamental module of our overall framework, and that it helps preserve useful frequency-domain information rather than indiscriminately suppressing high-frequency signals.
>
> 3. Q3
>
> We agree that a very short observation window may reduce the stability of frequency-domain representations and make the prediction task more challenging. To directly examine this issue, we conducted an additional experiment on Argoverse 2 (Table 4), where the TF-FACE input window is reduced from 50 frames to 20 frames, while the prediction horizon is fixed at 60 frames.
> Under this short-history setting, TF-FACE with 20→60 achieves minADE6 / minFDE6 / MR = 0.725/1.31/0.164. Notably, this is better than the strong baseline QCNet under the same 20→60 setting (0.747/1.35/0.174), and is also very close to QCNet with a much longer 50→60 history setting (0.726/1.29/0.167).
>
> These results suggest that the long-horizon prediction advantage of TF-FACE is not mainly dependent on longer history inputs, but rather arises from its adaptive and controllable use of frequency-domain information, which allows it to extract effective predictive cues even from short observation windows.
>
> On other weaknesses:
>
> We agree that TF-FACE is more complex than pure time-domain models. However, this additional complexity is necessary to address the limitations of pure time-domain models in explicitly disentangling low-frequency global trends and high-frequency local details. In other words, the added design is not introduced for complexity itself, but to solve the above modeling issue that conventional time-domain architectures cannot explicitly handle. In practice, as shown in Fig. 12, TF-FACE remains efficient, with 6.1M parameters and 37 ms average latency, while achieving the best accuracy among publicly available methods on the benchmark, thus providing the best overall balance among modeling capability, accuracy, and efficiency.
>
> We agree that in highly abrupt, short-interval motions, high-frequency details may become more entangled with global trends, and that the current manuscript discusses such cases insufficiently. However, TF-FACE does not rely on rigid frequency decoupling alone: GfreAttn adaptively reweights frequency components, and the high-frequency branch only provides auxiliary refinement. Moreover, abrupt motions are often influenced by preceding interaction dynamics, part of which can be captured from surrounding agents’ historical high-frequency signals.
>
> If accepted, we will make the code publicly available, and add all necessary supplementary materials in the camera-ready version to ensure full reproducibility.

---

> > ### Author Rebuttal · Reviewer_P8Gi · 2026-04-03
> >
> > Thank you for your detailed response. I will maintain my positive score.

---

> > > ### Author Response · Authors · 2026-04-07
> > >
> > > We sincerely thank the reviewer again for the careful reading, constructive feedback, and positive assessment of our work. We are particularly grateful that the reviewer found our previous rebuttal helpful and considered the concerns to be fully resolved.
> > >
> > > Beyond the initial rebuttal, here we would like to put more emphasis on the significance and originality of the proposed work. Trajectory prediction is a critical component of autonomous driving, as it directly affects downstream planning, safety, efficiency, and human-like interaction. Due to the strict space limitation of the initial rebuttal, we were not able to elaborate our motivation sufficiently. TF-FACE is motivated by two key issues: (1) low-frequency components that reflect motion trends and high-frequency components that capture motion details are coupled in time-domain signals, making it difficult to explicitly decouple and exploit global trends and local details; (2) the spectral bias of neural networks makes high-frequency motion patterns harder to model effectively. Accordingly, our framework is designed in a targeted manner: (1) a two-stage decoupling-coupling strategy is introduced to explicitly model low-frequency global trends and high-frequency local details; (2) a learnable GfreAttn is developed to adaptively adjust the weights of motion patterns across different frequency bands to better support modeling.
> > >
> > > We would also like to further clarify that our design is not a direct reuse of existing components, but a targeted integration tailored to the limitations of pure time-domain trajectory prediction methods in autonomous driving. In this sense, we believe TF-FACE provides a new frequency-aware modeling framework for trajectory prediction in autonomous driving. More broadly, we believe this line of research may also offer a useful perspective for future end-to-end autonomous driving systems, where frequency-aware modeling could be further explored in scene encoding, world modeling, behavior reasoning, and motion planning. This is one direction that we are highly interested in continuing to explore.
> > >
> > > We are sincerely encouraged by the reviewer’s positive feedback. If the reviewer feels that these additional clarifications further strengthen the significance and originality of our work, we would be very grateful if the score could be reconsidered accordingly.
> > >
> > > Thank you again for the reviewer’s time, support, and valuable suggestions.

---

### Official Review · Reviewer_3bJJ · 2026-03-12

**Soundness:** 3
**Presentation:** 3
**Significance:** 3
**Originality:** 3
**Overall Recommendation:** 5
**Confidence:** 4

**Summary:**

This paper proposes TF-FACE, a time–frequency fusion framework for multimodal trajectory prediction in autonomous driving. The key idea is to incorporate frequency-domain representations into trajectory modeling to better capture both long-term motion trends and short-term motion details.

The framework first encodes historical trajectories using a Frequency-domain Adaptive Fusion Encoding (FAE) module, which combines temporal convolution, a feature pyramid network, and a gated frequency-domain attention mechanism to adaptively enhance or suppress different spectral bands. A preliminary decoder then generates coarse future trajectories. These predictions are subsequently decomposed into low-frequency global trend features and high-frequency local motion features through dedicated modules. A dual-branch parallel decoder processes these components separately to generate global trend trajectories and local motion details, which are then combined to produce final multimodal predictions.

To further guide the learning process, the authors introduce a time–frequency dual-consistency loss, which applies frequency-aware supervision to different decoding stages and encourages consistency between predicted and ground-truth trajectories across spectral bands.

Experiments on Argoverse 1 and Argoverse 2 benchmarks demonstrate competitive or state-of-the-art performance across several trajectory prediction metrics, including Brier minFDE, minADE, and miss rate. Additional ablation studies and robustness evaluations suggest that the proposed frequency-domain modeling improves long-horizon trajectory prediction and enhances robustness to noisy inputs while maintaining real-time inference speed.

**Compliance With Llm Reviewing Policy:**

Affirmed.

**Final Justification:**

This paper proposes TF-FACE, a time–frequency fusion framework for trajectory prediction that integrates spectral representations with temporal modeling to better capture both global motion trends and local motion variations. The problem is important and well motivated, and the proposed approach provides a meaningful perspective by introducing frequency-domain reasoning into motion forecasting. The overall framework is technically well structured, with a clear separation between encoding, frequency decomposition, and dual-branch decoding.

From a soundness perspective, the method is technically solid and supported by comprehensive experiments on standard benchmarks such as Argoverse 1 and Argoverse 2. The ablation studies demonstrate the contribution of key components, and the evaluation across multiple metrics strengthens the empirical validity. The rebuttal further improves confidence by providing a detailed sensitivity analysis of the low-/high-frequency split, showing that the model is robust across a range of ratios and datasets. Additionally, the discussion of adaptive partitioning and its trade-off with computational cost is practical and well justified.

In terms of originality, while individual components (e.g., Fourier transforms, attention mechanisms) are not entirely new, the work presents a coherent and targeted integration of time–frequency modeling specifically tailored for trajectory prediction. The clarification provided in the rebuttal reinforces that the contribution lies in the structured design and application of frequency-aware modeling rather than isolated novelty of components.

Regarding significance, the work addresses a highly relevant problem in autonomous driving and demonstrates competitive performance, including improvements in long-horizon prediction and robustness. The additional experimental results beyond the original datasets (e.g., nuScenes) further strengthen the generalization claims. The idea of explicitly modeling frequency components in trajectory prediction is promising and could inspire further research in this direction.

For presentation, the paper is generally clear and well organized. The rebuttal indicates that the authors will further improve clarity by simplifying module descriptions and strengthening the related work discussion, which should enhance readability in the final version.

The rebuttal effectively addressed my main concerns, including sensitivity to the frequency split, computational efficiency, generalization, and interpretability of frequency components. While some aspects such as architectural complexity and reliance on heuristic design choices remain inherent to the approach, they are now sufficiently justified and do not undermine the overall contribution.

Overall, the rebuttal strengthens my confidence in the work and reinforces my initial assessment. The paper is technically sound, relevant, and offers a meaningful contribution to trajectory prediction research. I therefore maintain my recommendation of Accept.

**Key Questions For Authors:**

1.The split between low-frequency and high-frequency bands is determined empirically based on dataset statistics. Have the authors considered learning this frequency partition dynamically during training, and how sensitive is the model to this design choice across different datasets?

2.The architecture includes multiple frequency transforms (DFT/IDFT) and attention layers. Could the authors provide a clearer comparison of computational complexity and runtime against other recent trajectory prediction models?

3.The experiments focus mainly on the Argoverse datasets. How does the proposed framework perform on other motion forecasting benchmarks, such as Waymo Open Motion Dataset or nuScenes prediction benchmarks?

4.The paper argues that frequency-domain modeling improves long-horizon trajectory prediction. Can the authors provide additional analysis explaining which frequency components are most informative for different types of driving behaviors (e.g., lane changes, turns, or acceleration)?

**Limitations:**

Yes.

**Strengths And Weaknesses:**

Soundness

Strengths

The paper is technically well structured and the proposed framework is described with clear mathematical formulations. The integration of time-domain and frequency-domain representations is logically motivated by the observation that trajectory signals contain both low-frequency global trends and high-frequency motion details. The architecture introduces several coherent components, including frequency-domain attention, spectral feature extraction modules, and a dual-branch decoding mechanism.

The experimental evaluation is reasonably comprehensive. The method is evaluated on two widely used benchmarks (Argoverse 1 and Argoverse 2), and multiple metrics are reported to capture trajectory accuracy, multimodal reliability, and physical feasibility. Ablation studies demonstrate the contribution of key modules such as the gated frequency-domain attention mechanism and the frequency-specific feature extraction modules. The paper also evaluates robustness to input noise and investigates several hyperparameters, providing additional empirical insights.

Weaknesses

Despite the detailed design, some aspects of the method remain heuristic. In particular, the split between low-frequency and high-frequency bands is determined empirically based on dataset statistics rather than learned dynamically. This design choice may limit generalization across datasets or motion patterns. Additionally, although the paper motivates frequency-domain modeling conceptually, the theoretical justification for why the specific spectral operations improve trajectory prediction is limited.

The architecture also introduces a relatively large number of modules and transformations (DFT/IDFT operations, feature pyramid networks, multiple attention layers, and two-stage decoding). This increases implementation complexity and may make reproducibility more challenging. While the authors claim real-time inference capability, the runtime comparison with competing methods is relatively limited.

Presentation

Strengths

Overall, the paper is clearly organized and follows a logical progression from motivation to methodology and experiments. The figures illustrating the architecture and frequency-domain attention mechanism help readers understand the design of the proposed framework. The methodology section provides detailed equations and explanations for the core modules, and the appendices contain additional experimental details and implementation information.

The experiments are presented in a structured manner, including quantitative comparisons, ablation studies, robustness experiments, and qualitative visualizations. The inclusion of additional appendix analysis further improves transparency.

Weaknesses

The paper introduces many module names and abbreviations (e.g., FAE, GfreAttn, PreD, LFE, HFE, DuaD), which can make the architecture difficult to follow on first reading. Some sections of the introduction and methodology contain lengthy descriptions of module interactions that could be simplified or summarized more clearly. Additionally, the related work discussion could provide a deeper comparison with existing spectral learning approaches in sequence modeling.

Significance

Strengths

Trajectory prediction is a critical component of autonomous driving systems, and improving the modeling of long-term motion dynamics remains an important research problem. The proposed time–frequency modeling perspective provides an interesting alternative to purely time-domain approaches and may inspire further exploration of spectral representations in motion forecasting tasks.

The framework demonstrates competitive performance on established benchmarks and shows promising results in long-horizon prediction and robustness to noisy inputs. If validated further, frequency-domain modeling could become a useful complementary perspective for trajectory prediction research.

Weaknesses

Although the results are competitive, the performance improvements over recent state-of-the-art models are relatively modest. Additionally, the evaluation is limited primarily to the Argoverse datasets, and the broader impact of the approach across other trajectory prediction benchmarks or autonomous driving tasks remains unclear. As a result, the overall practical impact may be moderate rather than transformative.

Originality

Strengths

The paper proposes a relatively novel perspective by incorporating frequency-domain modeling throughout the trajectory prediction pipeline, including encoding, feature extraction, and decoding stages. The gated frequency-domain attention mechanism and the dual-branch decoding architecture provide an interesting design for separating global motion trends from local motion details.

The combination of spectral attention, frequency-based feature extraction, and time–frequency consistency losses represents a creative integration of ideas from signal processing and deep learning.

Weaknesses

While the integration is novel within trajectory prediction, the individual components (spectral attention, Fourier transforms in neural networks, and multi-scale modeling) have appeared in other sequence modeling contexts. Therefore, the contribution lies more in the systematic combination of these ideas rather than introducing an entirely new learning paradigm.

---

> ### Author Rebuttal · Authors · 2026-03-31
>
> We sincerely thank the reviewer for the careful, insightful, and constructive comments. We have carefully checked the issues raised and revised the manuscript accordingly.
>
> 1. Q1
>
> Specifically, when varying the low-/high-frequency split from 1:9 to 9:1:
> Brier-minFDE6: 1.519, 1.507, 1.504, 1.496, 1.499, 1.505, 1.515, 1.520, 1.523
> minFDE6: 0.908, 0.899, 0.896, 0.890, 0.892, 0.898, 0.906, 0.908, 0.910
>
> We have also explored adaptive frequency partition modules with dynamically learned boundaries. However, lightweight designs bring little benefit, while stronger variants increase the parameter count from 6.1M to 7.5M (+23%) and the average latency from 37 ms to 44 ms (+18%), with only about 1% improvement. Therefore, the current design provides a better practical trade-off between robustness and efficiency.
>
> Across Argoverse 1, Argoverse 2, and nuScenes, the 4:6 low/high-frequency split is the suitable choice. It provides a practical balance between preserving sufficient low-frequency trend and retaining enough high-frequency detail. More importantly, the model is not sensitive to the exact cut-off within a relatively coarse range. We believe this robustness mainly comes from: (1) GfreAttn adaptively reweights different frequency components; (2) the low-/high-frequency branches are coupled again in Stage II in a complementary manner, which further reduces reliance on a finely tuned manual partition. For more extreme datasets, one can further adopt the adaptive partition module we explored, or determine a suitable ratio with a lightweight engineering calibration process.
>
> 2. Q2
>
> We agree that TF-FACE is more complex than pure time-domain models. However, these designs are introduced specifically to address the limitation of conventional time-domain architectures, namely that they do not explicitly disentangle low-frequency global trends and high-frequency local details.
>
> In practice, TF-FACE remains lightweight, with 6.1M parameters, smaller than recent strong baselines, such as ProphNet (14.5M), Wayformer (11.2M), and QCNet (7.66M). Meanwhile, as shown in Fig. 12, TF-FACE achieves the best accuracy among publicly available benchmark methods with an average inference latency of only 37 ms. Therefore, TF-FACE still offers a strong overall balance among modeling capability, accuracy, and efficiency.
>
> 3. Q3
>
> In our early study, we also conducted experiments on nuScenes, where we obtained competitive results (minFDE1 6.67, minADE5 1.17). We compared several major motion forecasting benchmarks in terms of scale: Argoverse 1 contains 324k sequences, Argoverse 2 contains 250k scenarios, Waymo Open Motion Dataset contains 103k segments, and nuScenes prediction contains 32k training instances. Considering the larger scale and wider recognition of Argoverse 1/2 in recent trajectory prediction research, we finally chose them as the main evaluation platforms.
>
> 4. Q4
>
> We would like to clarify that specific driving behaviors do not have a strict one-to-one correspondence with fixed frequency bands. Instead, their frequency characteristics also depend on the interaction context, temporal duration, and motion intensity. Accordingly, TF-FACE does not manually assign behaviors to predefined frequency bands, but transforms motion behaviors into latent frequency-aware features through frequency decomposition and adaptive reweighting. In general, low-frequency components mainly capture global trends, while high-frequency components mainly contribute to local-detail refinement and short-term motion changes.
>
> On other weaknesses:
>
> TF-FACE is motivated by: (1) Low-frequency components that reflect trends and high-frequency components that capture details are coupled in time-domain signals, which cannot be decoupled for extracting global trends and local details; (2) the spectral bias of neural networks make high-frequency motion patterns harder to model effectively. Our framework is therefore designed in a targeted manner: (1) two-stage decoupling–coupling strategy to explicitly model low-frequency global trends and high-frequency local details; (2) learnable GfreAttn to adaptively adjust weights of motion patterns with different frequency bands to assist modeling.
>
> We will provide a clearer module correspondence table, simplify the description of module interactions, and summarize the overall architecture more clearly to improve readability. We will also strengthen the related-work discussion to better position our method with respect to existing spectral learning approaches.
>
> Finally, our design is not a direct reuse of existing components, but a targeted integration tailored to the limitations of autonomous driving trajectory prediction. In this sense, our contribution is to provide a new frequency-aware modeling framework for trajectory prediction in autonomous driving.
>
> If accepted, we will make the code publicly available and add all necessary supplementary materials in the camera-ready version to ensure full reproducibility.

---

> > ### Author Rebuttal · Reviewer_3bJJ · 2026-04-01
> >
> > Thank you for the detailed and constructive rebuttal. I appreciate the effort taken to address the concerns raised in the original review.
> >
> > The additional analysis on the low-/high-frequency split is helpful. The sensitivity study across different ratios, along with results on multiple datasets (including Argoverse 1/2 and nuScenes), improves confidence that the model is not overly sensitive to this design choice. The discussion on adaptive partitioning and its trade-off with computational cost is also reasonable and practically justified.
> >
> > The clarification regarding computational complexity and inference efficiency is also useful. Providing parameter counts and latency comparisons helps support the claim that the model remains lightweight despite its architectural complexity.
> >
> > I also appreciate the inclusion of additional experimental results beyond the originally reported datasets, which partially addresses concerns regarding generalization.
> >
> > The explanation regarding the interpretation of frequency components and the acknowledgment that behavior does not map directly to fixed frequency bands is fair and technically sound.
> >
> > Finally, the authors’ commitment to improving clarity, simplifying module descriptions, and strengthening the related work discussion is appreciated and should enhance the readability of the paper.
> >
> > Overall, the rebuttal addresses the main concerns raised in my review. The remaining issues are relatively minor and can be addressed in the final revision.

---

> > > ### Author Response · Authors · 2026-04-07
> > >
> > > We sincerely thank the reviewer for the careful follow-up evaluation and for the positive assessment of our rebuttal.
> > >
> > > We are very grateful that the reviewer finds the additional analyses, clarifications, and promised revisions helpful, and that the main concerns are considered addressed. We also appreciate the reviewer’s recognition of our efforts to improve the clarity, technical positioning, and overall presentation of the paper. We will carefully incorporate these suggestions in the final revision.

---

### Official Review · Reviewer_XN95 · 2026-03-12

**Soundness:** 2
**Presentation:** 2
**Significance:** 2
**Originality:** 2
**Overall Recommendation:** 4
**Confidence:** 3

**Summary:**

This paper addresses the problem of multimodal trajectory prediction for autonomous driving. The authors argue that existing time-domain methods insufficiently exploit latent frequency information, limiting their ability to capture long-term dependencies and short-term dynamics. To address this, they propose TF-FACE, a Time-Frequency Fusion learning framework with Frequency-domain Adaptive and Controllable Enhancement. The core method introduces a learnable gated frequency-domain attention mechanism within a Feature Pyramid Network for adaptive encoding, and a two-stage decoder with separate modules for extracting low-frequency global trends and high-frequency local details, guided by a novel band-specific time–frequency dual-consistency loss. Experiments on Argoverse 1 and 2 benchmarks demonstrate state-of-the-art or competitive performance across multiple metrics, with additional analyses showing improved robustness to input noise and real-time inference capability.

**Compliance With Llm Reviewing Policy:**

Affirmed.

**Final Justification:**

The authors have addressed several of my concerns: the formula typos, the duplicated definition of L_pre, and the incorrect QCNet citation are acknowledged and will be fixed in the revision.

**Key Questions For Authors:**

1) The paper repeatedly mentions that the low/high frequency division ratio is set based on dataset experience, and the optimal 4:6 ratio is given in Appendix Table 10. Please explain whether this threshold is obtained by tuning parameters on the validation set or fixed based on training set statistics? Were separate tunings performed on Argoverse 1 and Argoverse 2? If separate tunings were done, the persuasiveness of the method's cross-dataset generalization would be diminished.
2) High-frequency features usually represent abrupt changes and short-term maneuvering actions. If an agent has been moving smoothly in a straight line over the past 2 seconds, it will exhibit extremely weak high-frequency energy in the historical data. However, if it needs to make an emergency evasion in the next 3 seconds, strong high-frequency signals will be required in the future. Can directly reusing the historical high-frequency features guide the details of future abrupt changes? This design essentially implies the assumption that "high-frequency motion patterns have strong time translation invariance," but this is currently not reasonable enough in highly dynamic autonomous driving scenarios.
3) In equations (7) and (8), the shapes and acting dimensions of W_"Re" ,W_"Im" ,b_"Re" ,b_"Im"  are not clear enough, and the bias in the second equation seems to be written as b_"Re" . Please supplement with more explicit tensor dimension descriptions, and it would be better to provide pseudocode or implementation methods.
4) The QCNet entry in the references is not a paper on autonomous driving trajectory prediction. Please check the correctness of the citations for all core baselines, and confirm in the rebuttal which work the QCNet used in Tables 3/4 specifically refers to.

**Limitations:**

yes

**Strengths And Weaknesses:**

Strengths
1）The problem is clearly addressed, and the overall framework is complete.
2) This paper does not simply add a frequency-domain branch to a single module; instead, it embeds frequency-domain modeling into historical encoding, future trend extraction, local detail extraction, and loss design.  This integrated design of "encoding - decoding - training objectives" is the main contribution of this paper.
Weaknesses
1) In GfreAttn, the frequency division ratio "varies depending on the dataset", and the high-pass filtering threshold is also set based on experience. Although the method in the main text claims to be "adaptive and controllable", the key boundary between low and high frequencies itself is not learned but manually set. This makes "adaptivity" more reflected in the gating weights rather than the frequency band division itself.
2) In the second stage of decoding in the paper, low-frequency features are extracted from the future predicted trajectory Y_pre of the first stage, while high-frequency features are extracted from historical observed trajectories. The authors provide the reason that "future high-frequency components are difficult to directly use for prediction, and historical high-frequency components are more stable." However, there is a certain logical risk here. The extent to which "historical local high-frequency patterns" can be transferred to future local details depends on scenarios and behavior patterns. For example, future local changes such as sudden braking, abrupt lane changes, and interactions-triggered turns may not be fully recovered from historical high-frequency components.The paper does not specifically design experiments to test the boundaries of this assumption.
3) There are multiple errors in the formulas in the paper: In the freMLP formula of LFE, the second term of the bias seems to be written as b_Re, but it should actually be b_Im; In the frequency-domain loss formulas (15), (18), and (21) in the paper, "DDT" is written in multiple places, while "DFT" is consistently used throughout the rest of the paper, which appears to be a notation error. Additionally, the weight in formulas (15) and (18) is 1/(1+f^(-μ) ), and in formula (21) it is 1/(1+f^μ ), which is suspected to be a writing error; In formula (12), L_pre is defined as the sum of two terms. However, formula (13) redefines L_pre using the same symbol but introduces the Δ term, resulting in inconsistent formulas for the same variable, which causes confusion.
4) In the single-model results of Argoverse 1 in Table 1, TF-FACE and HPNet are tied in terms of Brier minFDE6, minADE6, and MR6. TF-FACE is slightly worse than HPNet by 0.01 in minFDE6, and only slightly better than DAC. Therefore, the claim of "state-of-the-art accuracy" is basically valid, but the advantage is not significant. It is more like competitive/on-par rather than being obviously superior.
5) The entry for QCNet in the references is clearly not the classic paper "Query-Centric Trajectory Prediction" in the field of autonomous driving trajectory prediction; instead, it resembles an object detection paper. If the QCNet results in Tables 3 and 4 are indeed from the QCNet in the field of trajectory prediction, then there is an obvious error in the current citation.

---

> ### Author Rebuttal · Authors · 2026-03-31
>
> We sincerely thank the reviewer for the careful, insightful, and constructive comments. We have carefully checked the issues raised and will revise the manuscript accordingly. We address each concern below.
>
> 1. Q1:
>
> As correctly pointed out, the adaptivity of our method mainly lies in the GfreAttn module, which dynamically reweights different frequency components, rather than explicitly learning a precise low/high-frequency boundary. We agree that the current version does not learn the cutoff in a fully end-to-end manner. Empirically, however, we observe that the model performance remains stable within a coarse and reasonable range of partition ratios, indicating low sensitivity to the exact boundary.
>
> Specifically, when varying the low/high-frequency split from 1:9 to 9:1:
>
> Brier_minFDE6: 1.519, 1.507, 1.504, 1.496, 1.499, 1.505, 1.515, 1.520, 1.523
>
> minFDE6: 0.908, 0.899, 0.896, 0.890, 0.892, 0.898, 0.906, 0.908, 0.910
>
> This robustness is largely due to GfreAttn, whose purpose is to reduce reliance on fine-grained partition tuning. We also note that this limitation has already been acknowledged in the conclusion. To further examine it, we implemented adaptive frequency partition modules with dynamically learned low/high-frequency boundaries. However, lightweight designs bring little benefit, while stronger variants increase the parameter count from 6.1M to 7.5M (+23%) and the average latency from 37 ms to 44 ms (+18%), with only about 1% improvement. Therefore, although we have explored this direction, the overall trade-off is unfavorable in practice, and we retain the current design for a better balance between robustness and efficiency. We will clarify this design choice and its limitation in the revised manuscript.
>
> 2. Q2:
>
> We agree that directly transferring historical high-frequency patterns to future behaviors may be unreliable in highly dynamic scenarios. However, our method does not assume a strict one-to-one transfer, nor does it rely on a strong time-translation invariance assumption. Instead, the historical high-frequency components of all traffic participants are jointly used as auxiliary cues for Stage II local-detail refinement, together with the future low-frequency trend estimated in Stage I. Therefore, the refined future local details are not determined solely by historical high-frequency information.
>
> From a broader perspective, most existing methods implicitly use full-band historical trajectories, including historical high-frequency components, for future prediction. In contrast, our method explicitly disentangles low-frequency trends and high-frequency details, and additionally uses future low-frequency information to guide trajectory prediction. We further conducted ablations on different high-frequency sources （Brier_minFDE6）: using historical HF gives 1.496, removing HF refinement gives 1.508, and using future HF gives 1.517. This suggests that future high-frequency components are less stable and may amplify prediction noise, while historical high-frequency signals provide more stable local-detail priors.
>
> Regarding the reviewer’s example, even if the target agent itself has weak historical high-frequency energy, its future abrupt behavior is often influenced by the surrounding traffic participants’ historical motion changes and interaction dynamics. Since our design leverages the historical high-frequency information of surrounding participants, it remains effective in most practical cases, and may become less effective only in highly extreme scenarios. We will make this limitation explicit in the revision.
>
> 3. Q3:
>
> To improve clarity, we will explicitly provide the implementation details of the frequency encoder as follows:
> Given $x \\in \\mathbb{R}^{B \\times D \\times L}$, we apply DFT to obtain $X_f \\in \\mathbb{C}^{B \\times F \\times D}$. Here, $B$ is the batch size, $D$ is the input dimension, $L$ is the sequence length, $F$ is the number of frequency bins, and $K$ is the embedding dimension.
> $$
> Y_f = X_f (W_{Re} + j W_{Im}) + (b_{Re} + j b_{Im}),
> $$ where $W_{Re}, W_{Im} \\in \\mathbb{R}^{D \\times K}$ and $b_{Re}, b_{Im} \\in \\mathbb{R}^{1 \\times 1 \\times K}$.
>
> 4. Q4:
>
> We replace the incorrect QCNet reference with the correct citation for Query-Centric Trajectory Prediction (Zhou et al., CVPR 2023).
>
> On other weaknesses:
>
> We have conducted a thorough check of the manuscript and have corrected all issues in the revision. We ensure these are presentation issues and do not affect the implementation or experimental results.
>
> We thank the reviewer for recognizing our results. Our method currently ranks first among publicly available models on the Argoverse benchmark. We also agree that the advantage is not large, and we will use more precise and cautious wording in the revision.
>
> If accepted, we will make the code publicly available and add all necessary supplementary materials in the camera-ready version to ensure full reproducibility.

---

> > ### Author Rebuttal · Reviewer_XN95 · 2026-04-08
> >
> > I thank the authors for their detailed and constructive rebuttal. The authors have addressed several of my concerns: the formula typos, the duplicated definition of L_pre, and the incorrect QCNet citation are acknowledged and will be fixed in the revision; the additional sensitivity sweep over the low/high-frequency split ratio and the clarified implementation details of the frequency encoder are also helpful. It is hoped that some of the wording can be appropriately adjusted. The low/high-frequency boundary is still set manually rather than learned, so I would suggest softening the "adaptive and controllable" wording accordingly. The reuse of historical high-frequency features for future local details is supported by the provided ablation on average, but a targeted analysis on hard cases would strengthen the claim. I would also encourage the authors to slightly soften the "state-of-the-art" wording on Argoverse 1, since on the main error metrics TF-FACE is essentially on par with the strongest single-model baselines.
> > Overall, I appreciate the authors' efforts and willingness to revise.

---

> > > ### Author Response · Authors · 2026-04-08
> > >
> > > We sincerely thank the reviewer for the careful follow-up evaluation and for the constructive and positive assessment. We are very grateful that the reviewer finds our rebuttal helpful and considers the main concerns to be adequately addressed. We also appreciate the reviewer’s valuable suggestions on further softening the wording regarding the adaptive frequency partition, the use of historical high-frequency features, and the “state-of-the-art” claim on Argoverse 1. We will carefully incorporate these suggestions in the revision to further improve the clarity, precision, and overall presentation of the paper.

---

### Decision · Program_Chairs · 2026-04-30

**Decision:**

Accept (regular)

**Comment:**

This paper proposes TF-FACE, a time–frequency fusion framework for multimodal trajectory prediction, integrating frequency-domain modeling across encoding, decoding, and training. It introduces gated frequency attention and a two-stage decoder for separating global trends and local details, achieving competitive results on Argoverse with robustness to noise.

Reviewers agree the paper is technically sound, well-structured, and well-motivated. Strengths include the holistic integration of frequency modeling, a coherent architecture, and comprehensive experiments with strong performance.

However, several concerns remain: the frequency band partition is manually designed rather than learned, weakening adaptivity; some modeling assumptions (e.g., reuse of high-frequency signals) lack full validation; and the architecture is relatively complex, with potential stability issues in the two-stage decoding.

The rebuttal improves clarity and provides additional evidence, but does not fully resolve these core issues.

Overall, these limitations concern design choices rather than fundamental flaws. Given the consistent empirical gains and positive reviewer consensus, I recommend acceptance.